# Uncertainty from choice of microphysics scheme in convection-permitting models significantly exceeds aerosol effects

Bethan White[1], Edward Gryspeerdt[2], Philip Stier[1], Hugh Morrison[3], Gregory Thompson[3], and Zak Kipling[4]

[1]Atmospheric, Oceanic and Planetary Physics, University of Oxford, Oxford, UK
[2]Institute for Meteorology, Universität Leipzig, Leipzig, Germany
[3]National Center for Atmospheric Research, Boulder, Colorado
[4]European Centre for Medium-Range Weather Forecasts, Shinfield Park, Reading, UK

*Correspondence to:* Bethan White (bethan.white@physics.ox.ac.uk)

**Abstract.** This study investigates the hydrometeor development and response to cloud droplet number concentration (CDNC) perturbations in convection-permitting model configurations. We present results from a real-data simulation of deep convection in the Congo basin, an idealised supercell case, and a warm-rain large-eddy simulation (LES). In each case we compare two frequently used double-moment bulk microphysics schemes and investigate the response to CDNC perturbations. We find that the variability among the two schemes, including the response to aerosol, differs widely between these cases. In all cases, differences in the simulated cloud morphology and precipitation are found to be significantly greater between the microphysics schemes than due to CDNC perturbations within each scheme. Further, we show that the response of the hydrometeors to CDNC perturbations differs strongly not just between microphysics schemes but also that the inter-scheme variability differs between cases of convection. Sensitivity tests show that the representation of autoconversion is the dominant factor that drives differences in rain production between the microphysics schemes in the idealised precipitating shallow cumulus case and in a sub-region of the Congo basin simulations dominated by liquid-phase processes. In this region, rain mass is also shown to be relatively insensitive to the radiative effects of an overlying layer of ice-phase cloud. The conversion of cloud ice to snow is the process responsible for differences in cold cloud bias between the schemes in the Congo. In the idealised supercell case, thermodynamic impacts on the storm system using different microphysics parameterisations can equal those due to aerosol effects. These results highlight the large uncertainty in cloud and precipitation responses to aerosol in convection-permitting simulations and have important implications not just for process studies of aerosol–convection interaction but also for global modelling studies of aerosol indirect effects. These results indicate the continuing need for tighter observational constraints of cloud processes and response to aerosol in a range of meteorological regimes.

## 1 Introduction

Deep convection has a significant influence on the state of the atmosphere and climate through shortwave and longwave radiative interactions, heat transfer through the release of latent heat, global heat redistribution, and plays an important part in the hydrological cycle through the conversion of water vapour to precipitation. One major way that aerosols can influence the

properties of deep convection is through their effect on cloud microphysics. By acting as cloud condensation nuclei (CCN), increased aerosol loading can lead to an increase in cloud droplet number concentration (CDNC) and subsequent reduction in cloud droplet size, which in turn has been hypothesised to suppress warm-phase precipitation (Albrecht, 1989). Some theoretical (e.g. Rosenfeld et al., 2008; Stevens and Feingold, 2009) and cloud (or cloud-system) resolving modelling studies (e.g. Fan et al., 2007b; Tao et al., 2007; Lebo and Seinfeld, 2011, amongst many others) have suggested that under certain conditions, precipitation suppression in the liquid phase may lead to an invigoration of deep convection and subsequent enhancement of convective precipitation. The detection of positive correlations between satellite-observed aerosol optical depth (AOD) and precipitation or convective cloud properties (e.g. Koren et al., 2005; Gryspeerdt et al., 2014) might suggest observational evidence of convective invigoration by aerosols. However, factors such as meteorological covariation and retrieval errors may contribute to or even dominate such correlations (Zhang et al., 2005; Mauger and Norris, 2007; Chand et al., 2012; Gryspeerdt et al., 2014). Complex process interactions in ice and mixed-phase microphysics, along with coupling to surface and radiative feedbacks and dynamics over a range of spatiotemporal scales, means that understanding and quantifying aerosol impacts on deep convection remains a significant challenge (e.g. Noppel et al., 2010; Seifert et al., 2012; Tao et al., 2012).

Representing cloud microphysical processes, which occur on length scales of microns to millimetres, has always been a significant challenge for atmospheric models. Even in cloud-resolving models, horizontal grid lengths tend to be on the order of kilometres to a few hundred metres at best, and so it is impossible for such models to explicitly simulate microphysical processes. There is a long history of microphysical parameterisation (see Khain et al., 2015, for a comprehensive review), and microphysics schemes today tend to fall into one of two categories: bin models, in which the size distribution of each hydrometeor class is explicitly calculated (e.g. Feingold et al., 1994; Stevens et al., 1996; Khain et al., 2004), and bulk models, in which a size distribution function typically is used to represent each hydrometeor class and one (or several) moments of the size distribution function are calculated explicitly (e.g. Kessler, 1969; Lin et al., 1983; Rutledge and Hobbs, 1983; Thompson et al., 2004; Morrison et al., 2005; Thompson et al., 2008, amongst many others). Bulk models are therefore very computationally efficient compared to bin models (often by at least two orders of magnitude, Jiang et al., 2000), and are used as standard in many atmospheric modelling systems today. Although certain aspects of cloud processes and aerosol indirect effects cannot be reproduced well in bulk schemes (see Khain et al., 2015, for a detailed analysis), there nevertheless remains a trade-off between how completely the hydrometeor size sprectra are represented and the physical domain size that can then be used in a simulation. For most applications, full bin microphysics (which can even resolve the autoconversion process of cloud water to rain) are only feasible using small domains and idealised simulations, which then cannot represent the dynamical feedbacks that can occur on larger domains (a notable exception, proving the cost of such simulations, are the multiple month-long case-study simulations using bin microphysics presented by Fan et al., 2013). Thus, studies using bulk and bin microphysics representations provide differently imperfect and thus complementary information. Indeed, bulk schemes remain as standard in global models, and there successful studies of aerosol indirect effects in global models have been performed using bulk microphysics (e.g. Zhang et al., 2016; Ghan et al., 2016).

Whilst early bulk microphysics schemes were single moment only (predicting only the k=1 moment of the particle size distribution equation, mass), a significant development has been predicting two moments of the size distribution equation (k=0, number concentration, and k=1, mass) (e.g. Meyers et al., 1997; Thompson et al., 2004; Morrison et al., 2005; Thompson et al., 2008), which has been shown to have improved results compared to single-moment schemes (e.g. Lynn and Khain, 2007; Morrison and Grabowski, 2007; Morrison et al., 2009; Kumjian and Ryzhkov, 2012; Saleeby and van den Heever, 2013). Indeed, although not widely used at present, three-moment schemes have been shown to further improve representations of large hail (Milbrandt and Yau, 2006; Loftus and Cotton, 2014) and precipitation reflectivities (Kumjian and Ryzhkov, 2012).

However, bulk schemes make a priori assumptions about the shape of the particle size distributions (usually approximated by exponential or gamma distributions, and more rarely by lognormal functions), whereas bin schemes calculate particle size distributions by solving explicit microphysical equations and make no a priori assumption about the particle size distribution shapes. This can lead to significant differences in the cloud and precipitation simulated by bin versus bulk schemes. For example, bulk schemes have been shown to underestimate areas of weak and stratiform rain in an MCS compared to a bin scheme which performed better against observations (Lynn et al., 2005a, b), and Li et al. (2009a, b) showed that a one-moment bulk scheme was shown to be worse at partitioning rain into stratiform and convective components in a continental squall line compared to a bin scheme (although many studies have showed that two-moment schemes are a significant improvement on single-moment schemes, Lynn and Khain (e.g. 2007); Morrison and Grabowski (e.g. 2007); Morrison et al. (e.g. 2009); Kumjian and Ryzhkov (e.g. 2012); Saleeby and van den Heever (e.g. 2013). Lynn and Khain (2007) found that, while all schemes overestimated maximum rain rates in a simulated MCS, all bulk schemes tested overpredicted average and maximum rain rates by a factor of 2 to 3, while bin schemes overestimated maximum rain rates by about 20%. In idealised supercell simulations, Khain and Lynn (2009) found that the Thompson et al. (2004) double-moment bulk scheme produced twice as much accumulated surface rain than a bin scheme, while Lebo et al. (2012) found that the Morrison et al. (2005) bulk scheme also produced twice as much surface rain as the same bin scheme used by Khain and Lynn (2009) in simulations of the same supercell. Investigations of the shape of the cloud droplet size distribution in large eddy simulations of non-precipitating shallow cumulus clouds with a bin (Igel and van den Heever, 2017a) and bulk (Igel and van den Heever, 2017b) scheme showed the importance of the cloud droplet size distribution shape parameter. In the bulk scheme, evaporation rates were much more sensitive to the value of the shape parameter than to the condensation rates, and thus the shape parameter strongly impacted cloud properties such as droplet number concentration, mean droplet diameter, and cloud fraction (Igel and van den Heever, 2017b). Bin scheme simulations suggested that the shape parameter should be based on the relationship between local values of the cloud droplet concentration and the relative width of the cloud droplet size distribution, rather than cloud mean values as are traditionally used (Igel and van den Heever, 2017a). Further, Igel and van den Heever (2017c) showed that despite other fundamental differences between the bin and bulk condensation parameterisations, differences in condensation rates could be predominantly explained by accounting for the width of the cloud droplet size distributions simulated by the bin scheme.

Seifert and Beheng (2006a) found that the most important factor in achieving agreement in concentrations and mass contents between bulk and bin schemes in simulations of continental and tropical maritime clouds was accurate representation of warm-phase autoconversion. Sensitivity tests of 4 different autoconversion parameterisations conducted by Fan et al. (2012a, b); Wang et al. (2013) showed that errors in predicting cloud water content in bulk schemes could be attributed to the saturation adjustment used in the calculation of evaporation and condensation. Likewise, Saleeby and van den Heever (2013) also showed, using 4 different types of autoconversion scheme, that saturation adjustment was the leading order factor in discrepancies of prediction of cloud water content by bulk schemes, and (Khain et al., 2016) found that tropical cyclones showed weak sensitivity to aerosol due to the use of saturation adjustment . In the ice-phase, Li et al. (2009a, b) found artificial spikes in heating rates from deposition and sublimation due to the saturation adjustment scheme. Bryan and Morrison (2012) found that even at very high resolution, convective cores in an idealized squall line simulation remained undiluted due to the saturation adjustment used in the bulk microphysics scheme. However, saturation adjustment alone is insufficient to explain all differences between bin and bulk schemes: in idealised supercell simulations using bulk microphysics both with saturation adjustment and without (where the scheme was modified to include an explicit representation of supersaturation predicted over each time step), Lebo et al. (2012) found that the use of saturation adjustment was able to explain differences between a bulk and bin scheme in the response of cold pool evolution and convective dynamics under polluted conditions, but was not sufficient to explain the large differences in the response of surface precipitation to aerosol loading.

Differences between bin and bulk schemes can often be traced to their different process representations. For example, some studies have found rain evaporation in bulk schemes to be too fast compared to bin schemes (Fan et al., 2012a, b; Wang et al., 2013; Li et al., 2009b; Shipway and Hill, 2012), bulk schemes have been found to have higher condensation and evaporation rates, but similar rates of freezing and melting, compared to bin schemes (Li et al., 2009a, b). Shipway and Hill (2012) compared rates of diffusional growth, collisions, sedimentation and surface precipitation in several bulk schemes against results from the Tel Aviv University bin scheme (Tzivion et al., 1987) and found that precipitation peaks in the bulk schemes were too sharp and too narrow compared to the bin scheme, whereas the bin scheme produced weaker precipitation covering an overall larger area than that in the bulk schemes. Morrison and Grabowski (2007) tested three different parameterisations of the coalescence process in the Morrison et al. (2005) bulk scheme against a bin scheme, under different aerosol loadings and in both warm stratocumulus and warm cumulus clouds, and found that for both the bulk and bin scheme, each representation of the coalescence process led to different averaged rain contents and mean raindrop diameters. Fan et al. (2013) showed that, because bulk schemes do not represent size-resolved ice particle fall speeds, they were unable compared to bin schemes to simulate the reduced fall velocities of ice and snow at upper levels from clean to polluted conditions in tropical, mid-latitude coastal and mid-latitude summertime inland continental deep convective clouds. Fan et al. (2013) also suggested that bulk schemes tended to artificially freeze large raindrops, due to the use of a fixed gamma distribution.

In some cases, tuning particular processes in bulk schemes has led to better agreement with bin schemes, e.g. tuning evaporation rates and fall velocities of graupel in single-moment bulk scheme simulations of a continental squall line (Li et al.,

2009a, b). Similarly, although no active tuning was performed, Seifert and Beheng (2006a) found that precipitation rates and accumulated precipitation values were in close agreement between simulations of continental and tropical maritime clouds in high and low CCN conditions using a bin and bulk scheme, with agreement between the bulk and bin scheme even greater in the high CCN case as compared to the low CCN case.

Not only do bin and bulk schemes often produce different results in terms of cloud and precipitation, but Fan et al. (2012a) found that the use of fixed CCN in a bulk scheme led to opposite CCN effects on convection and heavy rain compared to CCN effects when using a bin scheme. Similarly, Lebo and Seinfeld (2011) found an opposite response of accumulated surface rain to CCN in idealised supercell simulations using a bulk and bin scheme, and Khain and Lynn (2009) found a difference in the response of an idealised supercell to aerosol perturbations when a bin and bulk scheme was used, with the bulk scheme producing stronger updraughts and greater average precipitation than the bin scheme, and with the left-moving storm prevailing in the bulk simulation while the right-moving storm prevailed in the bin simulation. The differences were attributed to differences in the vertical velocities in the bin versus bulk schemes, which led to hydrometeors ascending to different altitudes with different directions of background flow.

Nevertheless, bulk schemes have shown sensitivity to aerosol. In simulations of tropical deep convection, Morrison and Grabowski (2011) found an ice-phase response to aerosol in which cloud top heights and anvil ice mixing ratios increase under polluted conditions, due to increased freezing of larger numbers of cloud droplets and subsequent higher ice particle concentrations with smaller sizes and reduced fall speeds. Indeed, a similar mechanism was later confirmed in bin-scheme simulations by Fan et al. (2013), who performed month-long simulations of deep convection over the tropical western Pacific, southeastern China and the U.S southern Great Plains. Further, Kalina et al. (2014) found that, autoconversion of cloud water to rain decreased under polluted conditions and, subsequently, near-surface rain and hail particles increased in size due to enhanced collection of cloud droplets. In simulations of deep convection over Florida using a bin-emulating bulk scheme, van den Heever et al. (2006) found that updraught strengths increased and anvil areas became smaller but better organized and with increased condensate mixing ratios. Similarly, in simulations of summertime convection over Germany using a two-moment bulk scheme, Seifert et al. (2012) found a strong aerosol effect on cloud properties such as condensate amounts and glaciation.

Unlike liquid cloud and rain drops (well-described by spheres of constant density), ice particles have a wide range of densities and shapes, making the representation of ice-phase microphysics in parameterisations much more difficult than the liquid phase. Traditionally, the approach in both bin (e.g. Khain et al., 2004) and bulk schemes (e.g. Meyers et al., 1997; Thompson et al., 2004; Morrison et al., 2005, etc.) was to partition ice particles into one of a fixed number of categories (e.g. cloud ice, snow, hail and graupel) each with its own specified density, shape distribution and physical parameters such as fall speeds. However, such partitioning oversimplifies the complex nature of ice-phase processes, requiring thresholds and parameters - often chosen on a relatively ad hoc basis - to determine the partitioning of ice particles into each category and for converting between categories. As such, it is unsurprising that simulations have been found to be highly sensitive to particle fall speeds and

densities (e.g. McFarquhar et al., 2006), the description of dense precipitating ice as hail or graupel categories (e.g. Morrison and Milbrandt, 2011; Bryan and Morrison, 2012), and changes in thresholds or rates for converting between ice categories (e.g. Morrison and Grabowski, 2008). Differences in ice-phase microphysics in bulk schemes have been shown to affect cloud biases especially at upper levels (Cintineo et al., 2014) and to affect ice–cloud–radiation feedbacks, with impacts on tropospheric

stability, triggering of deep convection, and surface precipitation (Hong et al., 2009). Such limitations have led to the development in more recent years of new representations of ice microphysics in bulk schemes, such as approaches which separately prognose ice mass mixing ratios grown by riming and vapour deposition (Morrison and Grabowski, 2008), where ice particle habit evolution is predicted by prognosing mixing ratios of ice crystal axes (Harrington et al., 2013), and where ice-phase particles are represented by several physical properties that evolve freely in time and space (Morrison and Milbrandt, 2015).

Although these developments are relatively new, they have already been shown to improve simulations of observed squall lines and orographic precipitation when compared to traditional two-moment bulk schemes (Morrison et al., 2015a).

Evaluations of microphysics schemes frequently involve comparison against observations of a real precipitation event (e.g. Morrison and Pinto, 2005). Often, multiple microphysics schemes are compared against each other and against observations

(e.g. Morrison and Pinto, 2006; Gallus Jr. and Pfeifer, 2008; Rajeevan et al., 2010; Jankov et al., 2011). Another common approach is to evaluate a single microphysics scheme against observations and then use different aerosol concentrations in the model to test the sensitivity of the observed storm to aerosol processes (e.g. van den Heever et al., 2006; Seifert et al., 2012). However, studies of different convective events in different regions, using different models with different microphysics schemes, often produce conflicting results on the nature of the storm repsonse to aerosol. Mesoscale studies of Florida con-

vection found that cloud water mass, updraught strength and surface precipitation tend to increase with increased aerosol concentration, while anvil areas decreased but contain greater condensate mass (van den Heever et al., 2006), whereas studies of summertime convective precipitation in Germany found that increased aerosol concentrations had a strong effect on cloud microphysical (and therefore radiative) properties but that the combined effects of microphysical and dynamical processes resulted in relatively little effect on surface precipitation (Seifert et al., 2012), similar to the findings of Thompson and Eid-

hammer (2014) in idealised and continental-scale simulations.

Detailed process modelling studies of aerosol–convection interactions often focus on the sensitivity of a single idealised model configuration (without large-scale meteorology or surface and radiative interactions) to perturbations, using either CCN spectra (e.g. Seifert and Beheng, 2006b; Morrison and Grabowski, 2011) or CDNC values (e.g. Thompson et al., 2004; Mor-

rison, 2012) as a proxy variable to test the sensitivity of the microphysics to aerosol. Many types of idealised models are used, ranging from flow over a 2D mountain (e.g. Thompson et al., 2004), 2D cloud-system resolving studies of interacting convective clouds (e.g. Morrison and Grabowski, 2011), to 3D simulations of idealised supercell storms (e.g. Khain and Lynn, 2009; Lebo and Seinfeld, 2011; Morrison, 2012; Lebo et al., 2012). With such a wide range of model configurations, convective and large-scale environments, microphysics parameterisations (bin and bulk models are both frequently used in idealised

studies of aerosol–convection interactions) and proxy variables used to represent aerosol processes, it is perhaps not surprising

that a consistent response of idealised convection to aerosol has not been seen and indeed, due to environment and regime-dependence, may not exist. Idealised flow over a 2D mountain using CDNC values to represent aerosol amounts showed that cloud water content increased with CDNC, and drizzle content decreased (Thompson et al., 2004), while a similar study using an idealised supercell configuration found that differences in the accumulated surface precipitation and convective mass flux

between polluted and pristine values of CDNC were very small (Morrison, 2012). In studies using modified CCN spectra to represent different levels of aerosol in a two-moment scheme, 2D ensemble simulations of interacting convective clouds have found that although cloud top heights and anvil ice increase under polluted conditions, convection actually weakens slightly compared to pristine conditions (Morrison and Grabowski, 2011). However, similar 3D simulations also using a two-moment microphysics scheme have shown that for isolated convective cells, increased aerosol leads to reduced total precipitation and

updraught velocity, while for multicell systems it leads to increased secondary convection, total precipitation and updraught velocities, whilst supercell systems are relatively insensitive to aerosol (Seifert and Beheng, 2006b). Additionally, environmental wind shear has been shown to have a role in determining the response of convective systems to aerosol, with increased aerosol loading invigorating convection under weak shear conditions and suppressing convection under strong shear in simulations performed with both bin (Fan et al., 2009) and bulk (Lebo and Morrison, 2014) microphysics schemes.

The focus of this work is to show within a single modelling framework that uncertainty on cloud impacts through choice of microphysics scheme can far exceed any aerosol effect seen within a single scheme, and that this is a consistent finding across different types of convection in different environments and types of simulation (all of which are known to impact the effect of aerosol loading on cloud development, e.g. Altaratz et al., 2014). Although we use two bulk microphysics schemes to

show this, there is a body of literature which identifies signals of aerosol impact on cloud in bulk schemes (e.g. Morrison and Grabowski, 2011; Morrison, 2012; Lebo et al., 2012; Kalina et al., 2014), albeit not always convective invigoration (see especially Lebo et al., 2012), and in bin-emulating bulk schemes (e.g. van den Heever et al., 2006; Lee and Feingold, 2010, 2013). Nevertheless, using a two-moment bulk scheme to simulate a single cumulonimbus in an environment characterised by high CAPE and low wind shear, Seifert and Beheng (2006a) found higher overshooting tops and larger sizes with increased aerosol

loading, indicating that in some environments bulk schemes are able to produce invigoration effects. In some cases, aerosol effects may be relatively small (less than 15%, e.g. Morrison and Grabowski, 2011). However, while some argue (fairly) that this at least in part is due to the limitations of bulk schemes to fully represent aerosol-cloud interactions (such as saturation adjustment, Lebo et al., 2012), others argue that this is consistent with the concept of clouds as a buffered system hypothesized by Stevens and Feingold (2009). Month-long simulations approaching the climatological scale using bin microphysics performed

by Fan et al. (2013) also showed aerosol impacts on precipitation on the order of a few %, however, those authors showed a significant aerosol impact on rain rates, rather than total rain amount, observing a shift towards heavier rain rates and fewer light rain rates under polluted conditions in two regions (a tropical environment and midlatitude coastal environment), although the response in a midlatitude inland summertime continental environment varied temporally over the simulation. Similarly to the environmental-dependence found by Fan et al. (2013), Kalina et al. (2014) showed that even in an idealised simulation of a

supercell using open boundaries and bulk microphysics, the relative humidity and shear used in the initial profile had an impact

on the aerosol effects observed in the simulation.

We perform high-resolution convection-permitting simulations with the Weather Research and Forecast (WRF) model in three configurations: a real-data simulation of deep convection in the Congo basin, an idealised supercell case, and a shallow convection large-eddy simulation (LES). In each case we compare hydrometeor development in two commonly used double-moment bulk schemes and investigate the response of each model configuration to CDNC perturbations. Our focus is not to provide a detailed process study of aerosol effects on convection per se (to do so in the context of multiple model configurations is beyond the scope of this paper), but rather to explore and identify uncertainty in the cloud and precipitation response to CDNC perturbations across a range of model configurations. We acknowledge that, due to a lack of fully coupled aerosol-cloud processes (e.g. supersaturation representation, droplet activation, wet deposition and buffering processes, Lebo et al., 2012; Stevens and Feingold, 2009; Lee and Feingold, 2010; Seifert et al., 2012), the magnitude of response of bulk microphysics schemes to CDNC perturbations may differ from that in schemes that explicitly treat cloud processing of aerosol. Our goal therefore is to highlight the large uncertainty in cloud and precipitation responses to perturbations of CDNC in convection-permitting models, even between multiple configurations of the same widely used model.

## 2 Experimental design

We use the Advanced Research WRF version 3 (Skamarock et al., 2008) in three different configurations: a real-data simulation of deep convection over the Congo basin, an idealised supercell simulation, and a warm-rain shallow cumulus LES simulation. WRF is a nonhydrostatic, compressible, 3D atmospheric model. We use version 3.5 of WRF in the Congo basin and the ide-alised supercell simulations, but version 3.3.1 was utilised for the warm-rain LES simulation because the LES packages were only available for this version of the model at this time (Yamaguchi and Feingold, 2012). In order to keep the simulations as consistent with each other as possible, we therefore implement the versions of the microphysics schemes from WRF version 3.5 into version 3.3.1 of the model for the LES simulations. Each set of simulations is performed using two microphysics pa-rameterisations, at three different prescribed CDNC values, resulting in a total of six simulations for each model configuration. The model configurations used in this study are summarised in Table 1.

### 2.1 Microphysics parameterisations

This study is presented as an indication of the uncertainty that can arise from choice of microphysics scheme alone, and thus we restrict our comparison to two double-moment bulk microphysics schemes rather than diversify into a comparison of bin schemes against bulk schemes. The literature surveyed in Section 1 indicates the wide range of differences that may be expected when comparing bulk against bin schemes. A significant body of work has shown that two-moment bulk microphysics schemes generally represent cloud and precipitation characteristics more realistically than single-moment schemes (most recently Mor-rison et al., 2009; Wu and Petty, 2010; Weverberg et al., 2013, 2014; Igel et al., 2015), and thus our study is restricted to the

comparison of two five-class, double-moment schemes commonly used in WRF and shown by Cintineo et al. (2014) to perform well against satellite observations of cloud in North America: that described by Morrison et al. (2005, 2009); Morrison and Milbrandt (2011) (hereafter Morrison, or abbreviated to MORR), and that described by Thompson et al. (2004, 2008) (hereafter Thompson, or THOM). Both schemes are two-moment in rain and ice (prognostic mass and number), while the Morrison scheme is also two-moment in snow and graupel. Both are single moment in cloud water: mass is the only prognostic liquid cloud variable, and CDNC is prescribed at a given value. Following the method used in many previous studies including that of Morrison (2012) we prescribe CDNC values (in this study, at 100, 250 and 2500 cm$^{-3}$) as a proxy for CCN varying under conditions ranging from clean to highly polluted. The list of microphysics configurations tested and the abbreviations used to describe them are summarised in Table 2.

## 2.2 Model configurations

The real-data Congo simulations use a model domain covering a 2100 km × 2100 km region over the Congo Basin (Figure 1), chosen due to the high frequency of isolated deep convective systems occurring in the region and also due to the presence of strong sources of biomass burning aerosol. The model initial and boundary conditions were generated from ERA-Interim reanalysis (Dee et al., 2011), starting at 00:00 UTC on 1 August 2007. The simulation start date was chosen to coincide with the onset of the seasonal peak in precipitation (Washington et al., 2013) and the simulation was integrated for 10 days (with a timestep of 12 s) in order to identify the nature of the convection and its repsonse to CDNC perturbations over timescales greater than that of the lifecycle of any individual convective system. We use a horizontal grid length of 4 km and 30 vertical levels with the standard WRF stretched vertical grid. This gives a vertical grid spacing of about 100 m in the lower levels, with grid spacing increasing towards the upper levels. Although 30 vertical levels may seem relatively coarse, it has been shown in a previous study to be sufficient to reproduce observed cloud morphology and resolve the vertical structure of aerosol and precipitation and their interactions in this region (Gryspeerdt et al., 2015). Longwave and shortwave radiation in the simulations are parameterised by the RRTM (Mlawer et al., 1997) and Goddard (Chou and Suarez, 1994) schemes, respectively. Other physics parameterisations (other than the microphysics schemes previously discussed) are the MM5 Monin-Obukhov similarity surface layer scheme available in WRF (which uses stability functions and surface fluxes from Dyer and Hicks, 1970; Paulson, 1970; Webb, 1970; Beljaars, 1994), the NOAH land surface model (Ek and Mahrt, 1991) and the YSU boundary layer scheme (Hong et al., 2006), also shown by Cintineo et al. (2014) to perform well.

The idealised supercell setup follows the standard 3D idealised supercell case available as part of the WRF modelling system. Boundary conditions are open on all lateral boundaries, and the model top and surface are free-slip. For consistency with the Congo basin simulations, we use a horizontal grid length of 4 km. The model domain is 1600 km × 1600 km in the horizontal and, for consistency with the Congo simulations, also uses 30 vertical levels with a model lid at 20 km. A Rayleigh damper with damping coefficient of 0.003 s$^{-1}$ is applied in the top 5 km of the model to prevent spurious wave reflection off the model top. Following the setup commonly used in idealised supercell studies (e.g. Morrison, 2012), surface energy fluxes,

surface drag, Coriolis acceleration and radiative transfer are neglected for simplicity, and the subgrid-scale horizontal and vertical mixing is calculated with a prognostic turbulent kinetic energy scheme (Skamarock et al., 2008). The model is initialized as in the idealised quarter-circle supercell test case available in WRF, using the analytic sounding of Weisman and Klemp (1982, 1984) and the quartercircle supercell hodograph of Weisman and Rotunno (2000) with the shear extended to a height of 7 km. Convection is triggered using a thermal perturbation in the centre of the domain, with maximum perturbation potential temperature of 3 K centred at a height of 1.5 km and with horizontal and vertical radii of 10 km and 1.5 km, respectively. All simulations are integrated for 2 h with a timestep of 12 s (the same timestep used in the Congo simulations).

The warm-rain shallow cumulus setup deviates from the other simulations in that it follows the LES intercomparison guidelines for the Precipitating Shallow Cumulus Case 1 (van Zanten et al., 2011) of the Rain in Shallow Cumulus Over the Ocean (RICO, Rauber et al., 2007) project and uses the RICO WRF LES package provided by Yamaguchi and Feingold (2012). The model domain is 12.8 km × 12.8 km × 4 km with a horizontal grid spacing of 100 m and uses 100 vertical levels, implying a vertical grid spacing of about 40 m. The lateral boundary conditions are doubly periodic. As in the idealised supercell simulations, surface energy fluxes, surface drag, Coriolis acceleration and radiative transfer are neglected for simplicity, and the subgrid-scale horizontal and vertical mixing is calculated with a prognostic TKE scheme. The surface conditions, wind and thermodynamic profiles, large-scale forcings and large-scale radiation, geostrophic wind, initial perturbations and translation velocity are prescribed following the RICO case guidelines (van Zanten et al., 2011). For consistency, we prescribe cloud droplet number concentrations at 100, 250 and 2500 $cm^{-3}$ following the other simulations in our study, instead of the 70 $cm^{-3}$ suggested for the standard RICO case. However, we also perform an extra simulation at 50 $cm^{-3}$. The simulations are integrated for 24 h with a timestep of 1 s.

## 3 Results

### 3.1 WRF Congo basin

Maps of simulated outgoing longwave radiation (OLR) and surface precipitation at 0700 UTC on 7 August 2007 (7 days into the simulation) indicate that the cloud morphological and precipitation differences for different microphysics schemes are much greater than the cloud and precipitation response within each scheme to different CDNC values (Figure 1). In the CONGO-MORR simulations, low OLR values (indicating cold, high cloud) are distributed across the domain. Precipitation at this time occurs only in cloud north of 3° S, but there is a large band of non-precipitating cold cloud across the south of the domain. There is little discernable response of the morphology of the OLR and precipitation in the CONGO-MORR simulations to different CDNC values (Figs. 1a, 1b and 1c). In comparison, cold cloud in the CONGO-THOM simulations ocurrs mostly north of 3° S (Figs. 1d, 1e and 1f). Less cloud forms in CONGO-THOM compared to CONGO-MORR, and the cloud generally has greater OLR values than that in CONGO-MORR. Some non-precipitating cloud occurs south of 3° S in the CONGO-THOM simulations, but the band is significantly weaker and warmer than in CONGO-MORR. The differences at this

snapshot are representative of differences that persist throughout the simulation. Frequency distributions of OLR over the entire 10-day simulation period show that CONGO-MORR has a much higher frequency of occurrence of colder, higher cloud (values about 120 W m$^2$) than CONGO-THOM (which increases in frequency slightly with increased CDNC), while CONGO-THOM has a much higher frequency of occurrence of warmer cloud (values about 270 W m$^2$) than CONGO-MORR (Figure 2a).

When compared to observations of OLR from the Geostationary Earth Radiation Budget (GERB, Harries et al., 2005) over the same region and period, CONGO-THOM represents warm cloud more consistently with GERB than CONGO-MORR, despite overpredicting colder cloud somewhat, while CONGO-MORR overpredicts higher cloud and underpredicts warm cloud compared to the observations (Figure 2a). However, despite a poorer prediction of cloud radiative properties, CONGO-MORR predicts surface precipitation better than CONGO-THOM when compared to observations from the Tropical Rainfall Mea-

suring Mission (TRMM, Huffman et al., 2007) merged product. Both schemes significantly overpredict surface precipitation compared to observations from the TRMM 3B42 product (although the spatial patterns of precipitation are reasonably similar), however total accumulated surface precipitation over the 10-day simulation period is much greater in CONGO-THOM than CONGO-MORR (Figure 3). Further differences are seen when the distributions of precipitation rates are compared, with CONGO-THOM overpredicting and CONGO-MORR underpredicting the occurrence of low precipitation rates compared to

TRMM, CONGO-MORR overpredicting and CONGO-THOM underpredicting moderate rates, and CONGO-THOM overpredicting the frequency of occurrence of very high precipitation rates (Figure 2b). That CONGO-MORR overpredicts the frequency of moderate rain rates and CONGO-THOM overpredicts the frequency of very high rain rates likely explains why both schemes overpredict total accumulated surface rain compared to the observations. Additionally, the overprediction of the frequency of very high precipitation rates by CONGO-THOM is likely the reason that the total accumulated surface precipita-

tion is much greater in this scheme than in CONGO-MORR (Figure 3a,b).

Further to the significant difference between the two schemes in their reproduction of cold cloud and precipitation rates, the updraught dynamics respond very differently to aerosol loading. Joint histograms of cloud top height in the convective updraughts and the radius of the updraughts show that the most significant dynamical difference between the simulations comes

from choice of microphysics scheme: the Morrison scheme has a tendency towards higher frequencies of wider updraught radii with higher cloud tops than the Thompson scheme (Figure S1 in the Supplement). Under increased values of CDNC, convection in the CONGO-MORR simulation shifts towards wider cores and higher core tops for mid-sized cores (radius 11 to 22 km), whilst there is a reduction in the frequency of smaller cores of all core top heights (Figure 2c). Conversely, convection under polluted conditions in the CONGO-THOM simulation shows a reduced frequency of occurrence of the highest

updraught cloud tops for all updraught radii under polluted conditions, with an increased frequency of occurrence of small updraught radii with lower cloud tops (Figure 2d). Therefore, a consistent aerosol response is observed in CONGO-THOM, resulting in smaller and lower convective updraughts (i.e. weakened convection under polluted conditions). Interestingly, both of these effects contradict the findings of Morrison and Grabowski (2011), who found an ice-phase response to aerosol in which cloud top heights and anvil ice mixing ratios increase under polluted conditions due to increased freezing of larger numbers of

cloud droplets and subsequent higher ice particle concentrations with smaller sizes and reduced fall speeds. However, we note

that we consider different values of CDNC / CCN to Morrison and Grabowski (2011) and that responses may be nonmonotonic Kalina et al. (2014), and that we consider a different case of convection (indeed, our 10-day Congo simulation covers many convective lifecycles). We note that the response of the convective updraughts to aerosol loading in these two bulk schemes cannot be attributed to saturation adjustment alone (the suggested effects of which on updraught invigoration are detailed in Khain et al., 2015), because both schemes use this method.

Not only does the simulated cloud and precipitation morphology differ significantly between microphysics schemes irrespective of the CDNC values used in the comparison, zonal-mean vertical sections of mass mixing ratios of the different hydrometeor classes show significant differences in the hydrometeor classes (due to microphysics) between CONGO-MORR and CONGO-THOM (Figure 4). The most significant difference between the two microphysics schemes is that south of 3° S, CONGO-MORR produces a large amount of high ice cloud between 300 and 150 hPa (Figs. 4a, b, c). Analysis of these vertical sections at hourly intervals throughout the simulation in conjunction with hourly maps of OLR as in Figure 1 show that this upper-level ice is transported from the convective anvils in the north of the domain to the non-convective region in the south of the domain (not shown). In comparison, CONGO-THOM produces significantly less ice, with almost no ice visible at this contour value (Figs. 4d, e, f). However, all three CONGO-THOM simulations form a large amount of non-precipitating low-level (950 to 850 hPa) liquid cloud south of 3° S. The bands of cloud seen south of 3° S in Figure 1 are therefore high ice cloud in the CONGO-MORR simulations and low liquid cloud in the CONGO-THOM simulations, illustrating not only a cloud morphological difference between the microphysics schemes but also a significant difference in the simulated hydrometeor classes and in the vertical distribution of hydrometeors. Even in the convective precipitating region in the north of the domain, the simulated hydrometeor classes differ significantly between the microphysics configurations, with the CONGO-MORR simulations generating more ice and less liquid cloud (Figs. 4a, b, c) and the CONGO-THOM simulations producing less ice and more liquid cloud (Figs. 4d, e, f). Rain is confined to the convective region in the north in CONGO-THOM, while in CONGO-MORR it is also present at low levels in the non-convective southern region of the domain which is dominated by liquid cloud in CONGO-THOM. We explain the mechanisms behind these differences later, but here we highlight that it is clear from Figure 4 that the differences in the simulated hydrometeors between microphysics schemes are much greater than the differences due to different levels of CDNC.

Because the partitioning of water into liquid and ice phases in the full-physics model configuration appears to depend strongly on the microphysics scheme, vertical sections of reflectivity occurrences derived from model hydrometeor fields passed through the Quickbeam radar simulator (Haynes et al., 2007) are compared against equivalent reflectivity occurrences from the CloudSat 2B-GEOPROF product (Marchand et al., 2008) (Figure 5). The histograms are derived from the reflectivity fields thresholded to include all values greater than -20 dBZ. The largest reflectivity values produced by the model occur in the convective region in the north of the domain, where the largest reflectivity values are detected by the satellite radar (Figure 5), also in agreement with the TRMM precipitation observations (Figure 3). However, both CONGO-MORR and CONGO-THOM have a large positive bias in reflectivity compared to the observations (Figure 5), indicative of limitations in the ability of both

bulk microphysics schemes to represent the observed vertical cloud structure in this geographic region over this time period. In general, CONGO-MORR has a much larger positive bias in reflectivity than CONGO-THOM (Figure 5). The CloudSat observations show a small frequency of occurrence of reflectivities detected at altitudes of 10 to 15 km in the south of the domain, which is well-represented by CONGO-THOM and indicates the overproduction of ice in CONGO-MORR (Figure 5).

Differences in the simulated hydrometeor classes between the schemes persist throughout the simulation and are illustrated by mean profiles of hydrometeor mass mixing ratios (Fig. 6). There is significantly more ice-phase condensate in the CONGO-M250 configuration (Fig. 6a), whereas the CONGO-T250 profile is dominated by a large amount of liquid cloud mass between the near-surface and 750 hPa (Fig. 6b). The differences in the total cloud water mass between the schemes are very large: at 950 hPa (the altitude with the greatest liquid cloud mass in CONGO-T250, Figure 6), cloud water mass contents are about 140× greater in CONGO-T250. The liquid cloud mass is always greater in CONGO-T250 than CONGO-M250 (Fig. 6c), by several orders of magnitude at some levels, but despite this the liquid phase does not appear to drive differences in precipitation between the microphysics schemes: CONGO-M250 has about 4× more rain mass in the mid-levels and 2× more rain mass near the surface than CONGO-T250 (Figs. 6a and b). In the ice phase, CONGO-M250 has only slightly more snow mass than CONGO-T250 but up to 10× more graupel mass (Figs. 6a and b) and while ice is a significant hydrometeor at upper levels in CONGO-M250, CONGO-T250 has almost no cloud ice at all (Figs. 6a and b). We note that the magnitude of the difference due to choice of scheme is the same when a bin scheme is used (Figures S2 and S3 in the Supplement).

Mean profiles over all condensed points (i.e. representing the mean values of each hydrometeor type, but not accounting for changes in absolute quantities across the model domain) show that CONGO-T250 has consistently more cloud water through the depth of the mean cloud compared to CONGO-M250 (Figures 6a, b), while CONGO-M250 produces more rain (Fig. 6a). That rain production in CONGO-M250 occurs mostly through the depths of the atmosphere where cloud water persists suggests that a significant proportion of the rain may be produced though autoconversion in CONGO-M250, although note that these mean cloud profiles are calculated over the entire domain and therefore incorporate both the deep convective region in the north and the warm-cloud region in the south, as seen in Fig. 4. Further, the two schemes show differences in the frozen hydrometeors, with the mean cloud in CONGO-M250 containing more graupel and less snow than CONGO-T250 (Fig. 6c). This may be an result of the use of distinct and different definitions of ice-phase hydrometeor categories in the two schemes, which have been shown to cause deficiencies in simulations of observed squall lines (Morrison and Milbrandt, 2015).

Not only does the partioning of ice amongst the hydrometeor classes differ between schemes, the response of the hydrometeors to CDNC perturbations also differs between schemes (Fig. 7). First note that the scale of the hydrometeor response to CDNC perturbations in the CONGO-MORR simulations is an order of magnitude smaller than the scale of the repsonse in the CONGO-THOM simulations. Over the entire domain, liquid cloud mass appears insensitive to CDNC perturbations in the CONGO-MORR configuration (Fig. 7a), although a reduction of mean-cloud liquid cloud mass under polluted conditions (Fig. 7c) indicates that there must be very few liquid cloud points in the CONGO-MORR simulation compared to

other hydrometeor types, notably ice (Fig. 7a). Very weak decreases in domain-mean near-surface rain mass may be evident under polluted conditions in CONGO-MORR, but this difference is on the order of $10^{-8}$ kg kg$^{-1}$ (Fig. 7a) A reduction in rain mass under polluted conditions is more evident in the mean rain profile (Fig. 7c), again indicating how few rainy points exist compared to other hydrometeor types in CONGO-MORR when considering the entire domain (Fig. 7a). Nearly all of the hydrometeor response in CONGO-MORR occurs in the ice phase processes: graupel mass decreases significantly under polluted conditions (Fig. 7a,c), while ice mass increases at upper levels in both a domain-mean and ice-mean sense (Fig. 7a,c). In contrast, the hydrometeor reponse to CDNC perturbations in the CONGO-THOM configuration is an order of magnitude greater than in CONGO-MORR and the dominant hydrometeor response to CDNC perturbations in CONGO-THOM occurs in the liquid phase. Not only does the CONGO-THOM configuration generate a significant amount more liquid cloud than the CONGO-MORR configuration (Figure 6c), but the liquid cloud mass increases under polluted conditions by an order of magnitude more than any other hydrometeor response (Fig. 7b,d). Rain mass is relatively insensitive to increased CDNC in CONGO-THOM (Fig. 7b,d). The significant difference between the response of the two schemes to perturbations in CDNC, with CONGO-MORR producing less liquid cloud and rain under polluted conditions while CONGO-THOM produces more cloud water, indicates significant differences in the cloud processes represented by the two schemes in this meteorological regime.

## 3.2  WRF idealised supercell

The results from the real-data Congo basin simulations indicate that the development of the simulated hydrometeor classes and the reponse of the hydrometeors to CDNC perturbations depend strongly on the choice of microphysics scheme. Although some previous studies have focused on the response of real-data case studies to both microphysics scheme and CDNC response (e.g. Fan et al., 2012a, 2013; Li et al., 2015), there is a much larger body of literature that investigates the repsonse of idealised supercell simulations to CDNC (or CCN) perturbations (e.g. Seifert and Beheng, 2006b; Khain and Lynn, 2009; Lebo and Seinfeld, 2011; Morrison, 2012). We therefore place our study in the wider context of the existing literature by investigating the reponse of a single isolated idealised supercell under both the MORR and THOM microphysics configurations to the same CDNC perturbations used in our Congo simulations, simultaneously allowing us to explore the case-dependence of the deep convective response to aerosol effects.

Figure 8 shows mean hydrometeor profiles from the idealised supercell model configurations under 'moderately polluted' prescribed CDNC values of 250 cm$^{-3}$. As in the Congo basin case, it is clear that the simulated hydrometeor classes differ significantly between schemes. In contrast to the Congo basin configuration, both the SUPER-MORR and SUPER-THOM configurations show similar behaviour in the liquid phase, producing similar profiles of liquid cloud mass and rain mass in both a domain-mean and hydrometeor class-mean sense (Figures 8a,d and 8b,e), and instead the most significant differences occur in the ice phase. Graupel dominates as the frozen precipitating hydrometeor in the SUPER-M250 configuration, amounting to about 4× the snow and ice masses at their peak amounts (Figure 8a,d). In contrast, snow is the dominant frozen precipitating

hydrometeor in the SUPER-T250 configuration, amounting to about 1.5× the graupel mass at their peak amounts and virtually no ice present (Figure 8b,e). Although there is very little difference between the SUPER-MORR and SUPER-THOM configurations in the liquid phase (except for the SUPER-MORR configuration producing about $2\times10^{-7}$ kg kg$^{-1}$ less domain-mean rain mass at the surface than SUPER-THOM, Fig. 8c), the SUPER-MORR configuration forms significantly more ice, more graupel and less snow than SUPER-THOM (highlighting that the partitioning of ice-phase hydrometeors into categories is very different, by design, in different microphysics schemes), and greater total quantities of frozen hydrometeors are present between 600 and about 150 hPa in SUPER-MORR compared to SUPER-THOM (Fig. 8c,f). This is a significant difference from the Congo real-data configuration, where the dominant contribution to the difference between the CONGO-MORR and CONGO-THOM configurations came from the liquid cloud (Figure 6c).

There is a more significant aerosol impact on hydrometeor mass in the supercell case than in the Congo case for both microphysics schemes, with mean responses over each hydrometeor type an order of magnitude greater in the supercell case (Fig. 7 compared to Fig. 9). Although many past studies have shown that aerosol impacts depend on cloud dynamics and thermodynamics (e.g. Khain and Lynn, 2009; Fan et al., 2009), we note that not only do the individual schemes respond differently to CDNC in different cases of convection (as expected) but that the way the schemes differ from each other in their response to CDNC is significantly different in the supercell case compared to the Congo case. The SUPER-MORR and SUPER-THOM cases differs qualitatively from the CONGO-MORR and CONGO-THOM cases, respectively, both in the altitudes at which the response occurs and the sign of the response of some of the hydrometeors. In the SUPER-MORR configuration, cloud water mass increases under polluted conditions, and rain mass is suppressed at mid-levels (between 600 and 450 hPa) but shows negligible response at the surface (Fig. 9a,c). In the ice phase, cloud ice increases under polluted conditions in SUPER-MORR, while graupel and snow decrease (Fig. 9a,c). Similarly, the hydrometeor response of the SUPER-THOM case to CDNC perturbations also differs in sign and in altitude to CONGO-THOM. In SUPER-THOM, cloud water mass increases and rain mass decreases under polluted conditions (Fig.9b,d), but unlike SUPER-MORR the decrease in rain is evident at the surface. Graupel mass decreases under polluted conditions in SUPER-THOM, similarly to SUPER-MORR but occurring over a much larger range of heights (Fig.9b,d), but unlike CONGO-THOM, which shows very little response to polluted conditions (Fig.7c). Interestingly, this is in contrast to Khain and Lynn (2009), who found an increase in graupel mass with increased CDNC in the Thompson scheme. However, their study was of 2D idealised squall line simulations, and considered CDNC values of 100, 500 and 100 drops per cm$^{-3}$. The dominant domain-mean response to increased CDNC perturbations in SUPER-THOM is an increase in snow mass between 550 and 150 hPa (Fig.9b), which likely comes from lofting of an increased mass of cloud water (Fig.9d). This is in contrast both to SUPER-MORR, where the dominant hydrometeor reponse occurred in the ice class (Fig.9a) despite and almost equal increase in lofted cloud water (Fig.9c), and to CONGO-THOM, where the dominant hydrometeor reponse occurred in the liquid cloud (Fig.7b). That both schemes show an increased lofting of cloud water under polluted conditions (Figs7c,d) but SUPER-MORR responds by generating more cloud ice (Figs7a,c) while SUPER-THOM shows an increase in snow (Figs7b,d) suggests differences in the processes that convert cloud ice to snow. This is explored later in Section 3.4. We emphasise that our main result shows that the variability due to microphysics scheme dominates any

aerosol impacts on microphysics. Results using the WRF-SBM in the idealised supercell case show that aerosol impacts in the bin scheme are of equal magnitude to those in the bulk schemes (Figure S4 in the Supplement).

To further investigate the importance of the difference in microphysics representations and the difference in their response to CDNC perturbations, Figure 10 includes the domain-mean total latent heating (sum of the latent heating from individual microphysical processes) contributions for each of the idealised supercell configurations. It can be seen that the choice of microphysics scheme can result in thermodynamic differences in the supercell system equal in magnitude to those arising from CDNC perturbations: between 500 and 250 hPa, the latent heating rate in the SUPER-M2500 configuration is almost identical to that in the SUPER-T250 configuration (solid red and dashed blue lines, Figure 10a). Thus, the magnitude and sign of the difference in the latent heating rate between SUPER-M250 and SUPER-T250 (blue solid and dashed lines, Figure 10a) is the same as that between SUPER-M2500 and SUPER-M250 (red and blue solid lines), and likewise the magnitude and sign of the difference in the latent heating rate between SUPER-M2500 and SUPER-T2500 (red solid and dashed lines) is the same as that between SUPER-T2500 and SUPER-T250 (red and blue dashed lines). In general, the SUPER-THOM configuration has a much stronger thermodynamic response to CDNC perturbations than the SUPER-MORR configuration, with latent heating rates consistently stronger throughout the atmosphere (Figure 10b). Overall, there is little evidence of convective invigoration (defined here as increases in upper-tropospheric heating, updraught strengths, cloud top height and surface precipitation) under increased CDNC values in either bulk microphysics scheme: although both schemes show increased latent heating in the upper troposphere and decreased heating at mid-levels under polluted conditions (Figure 10b), it has already been shown that there is no evidence of increased surface precipitation (Figure 9), and the upper tropospheric peak in latent heating can be seen to correspond to an increase in ice (SUPER-M250) or snow (SUPER-T250) at these levels (Figure 9). Neither is there any systematic or consistent evidence of increased mean updraught velocity in the convective cores (following the method of van den Heever et al., 2006; Lebo and Seinfeld, 2011) under polluted conditions (not shown), or in increased cloud top heights of the convective cores (Figs. 2c,d). This may not be surprising, as it has been suggested that bulk microphysics schemes are unable by design to produce convective updraught invigoration effects due to limitations in their representation of nucleation, sedimentation, and the way in which saturation adjustment limits diffusional growth (detailed in Khain and Lynn, 2009). Indeed, Lebo and Seinfeld (2011) found no latent heating effect of increased CCN in a bulk scheme used to simulate idealised deep convection, whereas with a bin scheme increased latent heating aloft was demonstrated. However, Lebo et al. (2012) found that saturation adjustment methods used in bulk schemes could explain differences in the response of cold pool evolution and convective dynamics between bin and bulk schemes to aerosol loading, but could not explain large differences in the response of surface precipitation. Further, some simulations using bulk schemes have identified invigoration-like effects under aerosol loading. For example, Lebo and Morrison (2014) found evidence of convective invigoration under increased aerosol loading in a bulk scheme under weak shear conditions (and suppressed convection under strong shear), similar to the findings of Fan et al. (2009) who found the same response in a bin scheme. Seifert and Beheng (2006a) also found higher overshooting tops and larger sizes of cumulonimbus in a weak shear environment with increased aerosol loading. Thus although our results agree with the body of the literature which doesn not identify a convective updraught invigoration effect when bulk microphysics

schemes are used, this is not necessarily attributable to the saturation adjustment method alone, and may also only hold for the particular convective environment (idealised supercell in strong shear) we consider.

### 3.3 WRF LES RICO

The results presented in Sections 3.1 and 3.2 indicate that not only is the way in which the schemes differ from each other not systematic between cases of convection, but that the difference between the response of the two schemes to CDNC across types of convection is also not systematic. The largest difference between the microphysics schemes in the real-data Congo basin simulations occurs in the liquid-phase hydrometeor development and response to CDNC. Making the assumption that the liquid phase is the first to respond to CDNC perturbations and the perturbation subsequently propagates to the ice phase, we

consider a case of precipitating shallow cumulus convection to investigate the liquid-phase differences between the schemes. Note that the 'baseline' hydrometeor profiles in Figure 11 show data from the configurations using a prescribed CDNC value of $100 \, \mathrm{cm}^{-3}$ (rather than the baseline value of $250 \, \mathrm{cm}^{-3}$ used in the Congo basin and idealised supercell deep convection cases in Figures 6 and 8), as this is more appropriate for a pristine marine environment. Even when we restrict our simulations to the liquid phase, differences in the simulated hydrometeor classes are evident. The dominant domain-mean difference between

the two schemes in the RICO case is clearly in the rain profile, with RICO-T100 producing significantly more rain than RICO-M100. Very little rain is present in the RICO-M100 configuration (Fig. 11a,d), whilst the RICO-T100 configuration produces a peak domain-mean rain mass of about $10^{-6} \, \mathrm{kg} \, \mathrm{kg}^{-1}$ (Fig. 11b). The liquid cloud profile is similar in both schemes, with RICO-M100 forming more cloud mass than RICO-T100 between 805 and 775 hPa in both the domain-mean and hydrometeor class-mean sense (Fig. 11c,f).

The response of the hydrometeors to CDNC perturbations also differs between schemes in the warm-rain RICO case (Fig. 12). In the RICO-MORR configuration, domain-mean rain and cloud mass both decrease under polluted conditions, although the rain response is very weak (on the order of $10^{-8} \, \mathrm{kg} \, \mathrm{kg}^{-1}$) and the dominant response is a reduction in liquid cloud mass (Fig. 12a). In contrast, a reduction in rain mass is the dominant hydrometeor response under polluted conditions

in the RICO-THOM configuration, and the decrease is nearly two orders of magnitude greater than that in RICO-MORR (Fig. 12b). The liquid cloud response to polluted conditions in RICO-THOM is weaker than the rain response, but still stronger than the cloud response in RICO-MORR. Cloud mass decreases under polluted conditions between 935 and 825 hPa, but increases at higher levels (Fig. 12b). Note that once again, the response of the simulated hydrometeors to CDNC perturbations differs between cases: under polluted conditions, RICO-MORR exhibits a decrease in cloud and rain mass, while CONGO-

MORR exhibits a decrease in rain mass with little reponse in the liquid cloud (Fig. 7a), and SUPER-MORR shows almost no liquid-phase response at all (Fig. 9a). Likewise, RICO-THOM exhibits a decrease in rain and increase in cloud mass under polluted conditions, while CONGO-THOM exhibits similar behaviour (Fig. 7b) but SUPER-THOM shows a decrease in rain mass with little response in the liquid cloud (Fig. 9b). When mean profiles of each hydrometeor class are considered, the two schemes actually show similar responses to CDNC (increased upper-level cloud mass and suppressed rain, Figs. 9c,d). This

indicates that the main response to CDNC in this case is not through the individual microphysical processes but through the absolute amounts of cloud and rain that are generated.

To illustrate the difference in the strength of response of the schemes to CDNC, total accumulated surface rain is shown for each RICO configuration in Figure 13a, along with an extra configuration using a 'very pristine' CDNC value of 50 cm$^{-3}$, and a series of sensitivity tests that will be discussed later. The 50 cm$^{-3}$ CDNC configuration has been added because even at a prescribed CDNC value of 100 cm$^{-3}$ very little rain production occurs in the RICO-MORR configuration. Warm rain formation differs strongly between schemes: very low CDNC values are required for the RICO-MORR configuration to produce any rain, whereas RICO-THOM produces significantly more rain at all CDNC values (Fig 13a). Even under very pristine conditions, the RICO-M50 configuration produces an order of magnitude less rain than RICO-T50 (Fig 13a). The different schemes also respond differently to CDNC perturbations. Rain production in RICO-MORR (which produces much less rain than RICO-THOM) shuts down very quickly as CDNC is increased: rain amounts are on the order of $10^2$ mm at a CDNC value of 50 cm$^{-3}$, $10^1$ mm at a CDNC value of 100 cm$^{-3}$, $10^{-1}$ mm at a CDNC value of 250 cm$^{-3}$, and rain production ceases completely at a CDNC value of 2500 cm$^{-3}$ (Fig 13a). In contrast, rain production persists for much larger CDNC values in RICO-THOM: rain amounts are on the order of $10^3$ mm at CDNC values of 50 cm$^{-3}$, $10^2$ mm at CDNC values of 100 cm$^{-3}$, $10^1$ mm at CDNC values of 250 cm$^{-3}$, and while rain amounts are very low at CDNC values of 2500 cm$^{-3}$ (on the order of $10^{-5}$ mm), rain production has not shut down completely (Fig 13a).

## 3.4 Sensitivity tests

Gilmore and Straka (2008) showed that the rain rates predicted by different autoconversion formulae in bulk schemes can vary by orders of magnitude. This sensitivity of results is also well highlighted in Thompson et al. (2004); note in particular their reference to Walko et al. (1995). Autoconversion is parameterised differently in the two microphysics schemes used in the current paper. The Thompson scheme follows an adaptation of Berry and Reinhardt (1974), while the Morrison scheme follows the method of Khairoutdinov and Kogan (2000).

Thompson et al. (2004) and Thompson et al. (2008) justify their choice of an adapted version of the Berry and Reinhardt (1974) autoconversion parameterisation through favourable comparison to results from the bin scheme of Geresdi (1998). Furthermore, implementation of Berry and Reinhardt (1974) in the the Thompson scheme begins collision-coalescence production of warm rain at almost exactly 14 microns. It is known that raindrop onset begins when the mean volume radius exceeds a critical value of 13 to 14 microns (Freud and Rosenfeld, 2012; Khain et al., 2013; Rosenfeld et al., 2014). This is one of the principle reasons the Berry and Reinhardt (1974) autoconversion scheme was chosen by Thompson et al. (2004, 2008) rather than Khairoutdinov and Kogan (2000).

While the Khairoutdinov and Kogan (2000) autoconversion scheme was initially developed and applied for LES of stratocumulus, other than varying the prescribed values of cloud droplet number concentrations we run the microphysics schemes in their baseline configurations. Thus, although we do not advocate the use of Khairoutdinov and Kogan (2000) for non-stratocumulus cases, the Morrison scheme is frequently used for simulations of deep convection. Similarly, as the Khairoutdinov and Kogan (2000) autoconversion scheme was developed for LES-scale studies, the authors recognise the potential importance of sub-grid cloud variability at the scales used in the present study. However, we note that we are running the model and microphysics schemes in the typical setup for a convection-permitting model (that is, neglecting sub-grid cloud variability), as one of the main aims of this study is to highlight uncertainty in commonly used model configurations which are exactly based on these schemes.

The autoconversion rates as a function of cloud water content for each of the model configurations are shown in Figure 14. Also shown is the cloud water content (up to the mean plus 2 standard deviations) of each configuration . It is immediately clear that the threshold cloud liquid content for autoconversion in the Morrison scheme (solid lines) is significantly lower than that in the Thompson scheme (dashed lines), i.e. rain production can occur at much lower cloud liquid water contents in Morrison. It is also clear from the mean, (mean+1 standard deviation) and (mean+2standard deviations) cloud water content limits that rain production through autoconversion ought to be possible in all model configurations. However, despite the higher cloud water content threshold for autoconversion in the Thompson scheme, autoconversion rates are much greater once the threshold is reached, and liquid cloud is converted to rain much faster in Thompson than in Morrison. From Figure 14, it appears that the threshold for autoconversion is unlikely to be reached very often in any of the T2500 cases. In the deep convective cases, rain can be generated through ice and mixed-phase processes, but in the RICO warm-rain case this cannot occur. This explains why, compared to more pristine conditions, cloud mass increases in RICO-T2500 while rain mass decreases (Figure 12b).

Because Figure 14 indicates that the autoconversion threshold may be at least in part responsible for this response in the RICO-THOM case, we replace the autoconversion parameterisation in the Morrison scheme with that from the Thompson scheme, and vice versa. We use the notation M100T to denote the Morrison microphysics scheme (at a CDNC value of $100 \, \mathrm{cm}^{-3}$) with Thompson autoconversion (that of Berry and Reinhardt, 1974), and T100M to denote the Thompson scheme with Morrison autoconversion (that of Khairoutdinov and Kogan, 2000). Differences in the domain-mean hydrometeor mixing ratio profiles for each of the autoconversion-swapped configurations in the RICO case are shown in Figure 15. It is immediately clear that, in the warm-rain configuration, simply swapping the autoconversion treatment makes the hydrometeor development of the microphysics schemes much more like each other. The difference between the RICO-M100 configuration with the Morrison and Thompson autoconversion parameterisations (Fig. 15a) is quantitatively and qualitatively very similar to the difference between the RICO-M100 and RICO-T100 configurations (Fig. 11c). Likewise, the difference between the RICO-T100 configuration with the Morrison and Thompson autoconversion parameterisations (Fig. 15b) and finally the difference between the RICO-T100 configuration with the Morrison autoconversion parameterisation and the RICO-M100 configuration with the Thompson autoconversion parameterisation (Fig. 15c) are also very similar to the difference between the RICO-M100 and

RICO-T100 configurations (Fig. 11c).

Similarly, swapping the autoconversion parameterisations between the microphysics schemes in the RICO cases makes the surface rain production of the microphysics schemes much more like each other. The accumulated surface rainfall in the RICO-M100T configuration looks much more similar to the surface rainfall in the RICO-T100 configuration than it does to the RICO-M100 configuration (Fig. 13a). Rain amounts are on the order of $10^2$ mm in RICO-M100T and RICO-T100, whereas in RICO-M100 it is two orders of magnitude smaller (Fig. 13a). Likewise, the accumulated surface rainfall in the RICO-T100M configuration is of order $10^1$ mm compared to $10^2$ mm in the standard RICO-T100 case (Fig. 13a). To further test the importance of autoconversion in the liquid phase simulations, we first turn off autoconversion completely in the 100 cm$^{-3}$ CDNC simulations, and then allow autoconversion to occur but prevent the accretion of cloud water by rain. By design, in the absence of ice processes no precipitation occurs without autoconversion of cloud water to rain (Fig.13a, M100noAUTO and T100noAUTO). However, in the RICO 100 cm$^{-3}$ CDNC liquid-phase configuration the Thompson scheme can produce surface rain from autoconversion alone (albeit two orders of magnitude less than when rain can also accrete cloud water, (Fig. 13a, T100noACCR and T100), showing that autoconversion acts almost like a "trigger" in this scheme, after which accretion takes over the rain production process. (Indeed, in nearly all schemes, rain formation from accretion (once triggered) is orders of magnitude larger than from autoconversion.) In contrast, zero surface precipitation is produced in RICO M100noACCR (Fig. 13a), showing that in this (liquid-phase only) configuration the Morrison scheme requires both the autoconversion of cloud droplets to rain and the accretion of rain by cloud droplets in order to produce surface precipitation.

Despite the significant effect of autoconversion in the liquid-phase simulations, changing the autoconversion parameterisation in the idealised supercell case has very little effect on the hydrometeor development (results not shown). This is unsurprising, as ice and mixed-phase processes will dominate this shear-driven deep convective environment. However, the Congo basin configurations show large differences between microphysics schemes in the partioning of water into liquid and ice phases (CONGO-THOM produces much more liquid cloud; CONGO-MORR produces much more ice). In the CONGO-THOM configurations, the liquid-phase response to increased CDNC is also very similar to the RICO-THOM response (increased liquid cloud mass and decreased rain mass, Fig. 7b). When the Thompson autoconversion treatment is implemented in the Morrison scheme, rain production in the southern half of the domain ceases in CONGO-M250T, and the liquid phase is instead represented by low-level cloud with structure similar to the CONGO-T250 configuration (Figure 16a, compared to Figure 4e). To test if radiative effects associated with large amounts of anvil ice drive or contribute to the differences in low cloud, we also set the ice extinction coefficient to zero in both the longwave and shortwave radiation schemes in CONGO-M250. However, this has no effect on the low-cloud characteristics (Figure 16e compared to Figure 4b), and we therefore conclude that autoconversion of cloud water to rain is the factor dominating the absence of low-level cloud in the south of the domain in the CONGO-MORR simulations. In contrast, autoconversion is a less significant process in the CONGO-T250 configuration. Implementing the Morrison autoconversion treatment in the Thompson scheme has very little effect on the hydrometeor structure in the CONGO-T250M configuration compared to the CONGO-M250 configuration (Figure 16b, compared to Figure 4b). As a

final test, the autoconversion process is turned off in both of the microphysics schemes. This confirms that autoconversion dominates the lack of low cloud in CONGO-M250: the resulting liquid-phase hydrometeor structure (Figure 16c) is similar to both CONGO-T250 (Fig. 7b) and CONGO-M250T (Figure 16a). This also confirms that autoconversion is much less significant in the CONGO-THOM configurations: the bulk hydrometeor structure when autoconversion is turned off in CONGO-T250 (Figure 16d) is very similar to both CONGO-T250 (Fig. 7b) and CONGO-T250M (Figure 16b).

We have shown that the autoconversion process is responsible for the removal of the large cloud mass at low levels in the model configuration with the Morrison microphysics scheme. We also see that this low-level liquid phase cloud mass forms when we run the same simulation using the WRF bin microphysics implementation (the SBM part of the Hebrew University Cloud Model, Khain et al., 2011), although to a lesser extent than in the Thompson simulations, and the warm cloud produced by the WRF-SBM produces rain (Figs. S2 and S3 in the Supplement). We therefore suggest that it is not the Thompson scheme per se which is responsible for producing the low-level cloud mass, but rather the larger-scale meteorological conditions present in which these simulations are performed.

A further significant difference between the two schemes in the Congo simulations is the generation of large amounts of upper-level ice in CONGO-M250 which is not present in CONGO-T250 (Figs. 4b,e). In the Thompson scheme, the fraction of ice mass with a diameter greater than 125 $\mu$m is instantaneously transferred into the snow category (Thompson et al., 2008). The same threshold size for cloud ice autoconversion to snow is used in the Morrison scheme, but the process is parameterised differently (Morrison et al., 2005). Because the Morrison scheme appears to produce large amounts of anvil cloudiness for the Congo case, which is not seen in the observations (Figure 5), we perform further sensitivity tests in which we reduce the threshold size for cloud ice autoconversion in the Morrison scheme to 50% of its original value (Figure 16f), 10% of its original value (Figure 16g), and then finally replace the autoconversion of cloud ice to snow in the Morrison scheme with the parameterisation used in the Thompson scheme (Figure 16h). In all tests, the upper-level anvil ice is reduced significantly. Using the lowest value of the threshold size for cloud ice autoconversion reduces the anvil cloud, because almost all of the ice is immediately converted to snow (Figure 16g). However, using the Thompson ice autoconversion representation in the Morrison scheme significantly reduces the amount of cloud ice in the simulation, and all of the detrained anvil ice is removed (Figure 16h). This suggests that for the particular Congo simulation we have investigated, the conversion of cloud ice to snow is the main factor leading to the significant difference in anvil cloudiness between the two schemes, and is responsible for the difference in upper-level cloud between the CONGO-M250 simulation and the observations (Figure 5). Indeed, we note that in equivalent simulations performed with the WRF-SBM, the same persistent upper-level ice forms (Figure S2 in the Supplement). This shows that differences resulting from conversion of one ice category into another is a limitation of any scheme whether bin or bulk which uses fixed ice categories. Our results provide further evidence that the use of discrete ice-phase hydrometeor categories may be detrimental to the correct simulation of cloud, and suggests that new schemes which do not use such partitioning may give better results (e.g. Morrison and Grabowski, 2008; Harrington et al., 2013; Morrison et al., 2015b).

Our results show little impact of aerosol on precipitation in the Congo basin (Fig. 2b, Fig. 7), which is also seen when considering total accumulated surface precipitation (Fig. 13b), although CONGO-T2500 exhibits weak preciptation suppression under polluted conditions). This may be due to the longer duration of these simulations, performed over a larger domain, allowing the interaction of many cloud systems rather than considering the lifetime of a single isolated cloud. However, we
also see that although the representation of autoconversion has a significant effect on the vertical hydrometeor structure in the CONGO-M250 configurations (Fig. 4b, Fig. 16a and Fig. 16c), it has a much weaker effect on total surface precipitation (Fig. 13b, M250, T250, M250T, T250M, M250noAUTO and T250noAUTO). This is perhaps unsurprising, as the dominant contribution to the accumulated surface precipitation over the Congo domain will be from ice processes in the convective region and not from the liquid-phase cloud. Although the lack of impact of aerosol on precipitation in the Congo simulations
may be due to the use of bulk schemes in this study for the reasons detailed in Khain et al. (2015), and perhaps a different response would be seen using a bin scheme (e.g. Khain and Lynn, 2009; Lebo and Seinfeld, 2011), other studies using bulk and bin-bulk schemes have identified aerosol impacts on precipitation of up to about 15% (e.g. van den Heever et al., 2006; Lee and Feingold, 2010; Lee, 2012; Morrison and Grabowski, 2011; Morrison, 2012; Lebo et al., 2012; Lee and Feingold, 2013; Kalina et al., 2014). Indeed, even studies using bin schemes have been shown to have little impact on total precipitation,
although inducing a shift in rainfall rates (Fan et al., 2013). Therefore, we note again that the choice of microphysics scheme, rather than aerosol response in either scheme, is the dominant contribution to uncertainty in the total precipitation.

## 4  Discussion and Conclusions

This study considered the cloud and precipitation development using two double-moment bulk microphysics schemes (Morri-
son et al., 2009; Thompson et al., 2008) to perform cloud system-resolving simulations of 3 types of convection, two of which were idealised (one deep convection case with open boundaries and one shallow cumulus case with periodic boundaries), and one real-data case of deep convection in the Congo basin using meteorological initial and boundary conditions. We tested the sensitivity of the simulated hydrometeors and precipitation to the microphysics scheme and to CDNC perturbations. The simulations were performed to explore the uncertainty in cloud and precipitation development and response to aerosol perturbations
in convection-permitting models that can arise from the microphysics representation. We find that the variability among the two schemes, including the response to aerosol, differs widely between these cases. Although previous studies have found large sensitivity to choice of microphysics schemes (e.g. Khain et al., 2015, 2016), we show this in a consistent setup by considering different cases with the same model and same CDNC values and constraining as many other possible sources of variability as is feasible. Our results show that for the bulk schemes used in these simulations, aerosol effects are dominated by the uncertainty
in cloud and precipitation development which arises from the choice of microphysics scheme. This result was true for multiple cloud types in multiple environmental conditions.

A key finding is that the difference between the two schemes, including their response to CDNC, in different environments and cloud types is not systematic. This could perhaps be related to the nonmonotonic response to aerosol in different environments found by Kalina et al. (2014) (although their study only considered simulations of idealised supercells with a single bulk scheme and four environmental soundings). This nonmonotonic response was attributed to compensating changes in the microphysical processes under polluted conditions.

The maximum relative difference in mass mixing ratio between each hydrometeor class in the M250 and T250 configurations for each case of convection is summarised in Table 3. Not only are the maximum differences in the domain-mean profiles of the hydrometeor classes simulated by each microphysics scheme on the order of at least tens of percent, but it is also clear that both the magnitude and sign of the difference varies between cases. In some cases, the magnitude of the difference is huge: most notably, in the Congo basin case the maximum difference in liquid cloud mass between the Morrison and Thompson schemes is on the order of $10^4$ kg kg$^{-1}$ more in Thompson (whereas in the RICO shallow cumulus case the maximum difference is on the order of $10^1$ kg kg$^{-1}$ less in Thompson). Likewise, in the RICO case the maximum difference in rain mass between the Morrison and Thompson schemes is on the order of $10^3$ kg kg$^{-1}$ more in Thompson (whereas in the Congo basin case the maximum difference is on the order of $10^1$ kg kg$^{-1}$ less in Thompson). Even for hydrometeors that have differences of the same order of magnitude, the sign of the difference can vary between cases. This result highlights the need for better observational constraints on mixed-phase and ice cloud microphysics and hydrometeors, and also perhaps the need for a shift in the development of microphysics parameterisations away from schemes which (somewhat arbitrarily) partition hydrometeors into separate categories. This is also supported by our sensitivity tests of autoconversion of cloud ice to snow in our Congo simulations.

Another key finding is that the cloud morphological difference and the difference in the hydrometeors between different schemes is significantly larger than that due to CDNC perturbations. Although we have restricted our study to the comparison of double-moment bulk microphysics schemes, this result is consistent with Khain and Lynn (2009), who found that the difference in convection between a bulk and a bin scheme was much greater than the difference within each scheme to varying aerosol concentrations. Some studies have found a significantly weaker response to aerosol when using bulk schemes compared to bin schemes, e.g. Khain and Lynn (2009); Lebo and Seinfeld (2011). In idealised simulations of continental deep convection, Lebo and Seinfeld (2011) found that increases in CCN concentrations led to increased ice mass and total condensed water mass aloft in both bin and bulk schemes, but increased surface domain-averaged cumulative surface precipitation in the bulk scheme compared to a decrease in the bin scheme. This was shown to be because the relative increase in condensate mass aloft under polluted conditions was found to be much larger in the simulations performed with bulk microphysics, a result of increased numbers of smaller cloud particles with slower sedimentation speeds, thus resulting in reduced surface precipitation. However, in our idealised supercell simulations we find a similar magnitude of response to aerosol when using a bin scheme as the response in the two bulk schemes which are the focus of this study.

That cloud and precipitation development and their aerosol response differs across different cloud types in different large-scale environments is expected. Many studies have shown that aerosol effects on precipitation depend on the large-scale environment and cloud type (e.g. Khain et al., 2004; Fan et al., 2007b; Lynn et al., 2005a, b; Lynn and Khain, 2007; Seifert and Beheng, 2006a; Tao et al., 2007), for reasons related to differences in different cloud types between the timescale of increased

sedimentation through aerosol loading and subsequent sublimation and evaporation timescales. Further, several studies of deep convection have found that the effects of aerosol on deep convection are much weaker than that of relative humidity (e.g. van den Heever et al., 2006; Fan et al., 2007a; Khain and Lynn, 2009). Fan et al. (2007b) found that in idealised simulations of continental and maritime clouds using bin microphysics the magnitude and even the sign of aerosol effects on precipitation depended on relative humidity. Fan et al. (2007a) found that aerosol response in idealised simulations of clouds using bin

microphysics and soundings from Houston, Texas strongly depended on relative humidity, having negligible effect on cloud properties and precipitation in dry air but more significant effects in humid air. Conversely, in idealised low-precipitation supercell simulations with bulk microphysics and dry low-level humidity performed as part of the study by Kalina et al. 2014, cold pool area decreased by 84% and domain-averaged precipitation reduced by 50% under polluted conditions whereas it was insensitive to polluted conditions when a moist sounding was used. Thus, assuming that the response in our simulations would

likely be more similar to the results found for bulk microphysics by Kalina et al. (2014) the magnitude of our results in the supercell case (which uses a moist sounding) may be smaller than it would be in drier environmental conditions.

In 10 day simulations of deep convection in the Congo basin in August 2007, we find that both the Morrison and Thompson schemes have a significant positive bias in cloud and surface precipitation compared to GERB and TRMM. This may be in

part attributable to the positive moist bias in the Congo basin in the ERA-Interim reanalysis (used as boundary data for the Congo simulation) when compared to other reanalyses (Washington et al., 2013). Despite the positive cloud fraction bias in both schemes, we find that the Thompson scheme compares better than the Morrison scheme against observed cloud fractions, largely due to the overproduction of upper-level ice in the Morrison scheme. This is in agreement with Cintineo et al. (2014), who found that (despite the two schemes having different biases at different levels) the Thompson scheme outperformed the

Morrison scheme overall against satellite observations of cloud in North America due to its more accurate upper-level cloud distribution, whereas the Morrison scheme had too much upper-level cloud through overproduction of ice. This bias is attributable to differences in the way in which the two schemes convert cloud ice to snow. However, we also find that despite a positive surface precipitation bias in both schemes, the Morrison scheme compares better to observations in this region over this period. Morrison and Grabowski (2007) found that differences in accumulated precipitation produced by warm stratocu-

mulus and warm cumulus clouds using different microphysics schemes were only on the order of 10 to 20%, suggesting that accumulated rain is largely controlled by large-scale atmospheric properties. However, differences in accumulated rain in our Congo simulations can be attributed to differences in the microphysics schemes, because all simulations used the same input and boundary data and therefore are under the influence of the same large-scale atmospheric conditions. That one scheme best represents cold cloud compared to observations but the other scheme better reproduces accumulated precipitation makes it

difficult to conclude that one scheme outperforms another overall, and suggests that when setting up a model configuration for

research purposes, one consideration to distinguish between the use of these two particular schemes may be whether surface precipitation or radiative effects are more important to the research question.

We note here that the RRTM LW and Goddard SW radiation schemes used in these simulations are only coupled to the microphysics through the hydrometeor masses and not the numbers. This coupling therefore cannot account for changes in hydrometeor sizes, and thus some aerosol effects will be missing from these simulations. Additionally, the microphysics-radiation coupling is only through cloud water and ice, and none of the other frozen species. This missing aerosol effect may have an especially important impact in our Congo simulations, where the Morrison scheme develops and retains significant amounts of upper-level ice, whereas the Thompson scheme converts nearly all the ice to snow, which the radiation scheme will not see. This could have significant radiative flux and feedback impacts (Thompson et al., 2016), which in itself originates from the use of somewhat arbitrarily defined ice categories (e.g. if the size parameter at which cloud ice is converted to snow is changed, a bulk mass of cloud ice is removed from the radiatively-coupled ice category and moved into the non-radiatively coupled snow category).

We present the new result that variabilty in aerosol response due to choice of microphysics scheme differs not just between schemes, but that the inter-scheme variability differs between cases of convection. The maximum relative difference in the domain-mean hydrometeor profiles between polluted and pristine CDNC values for each of the model configurations is summarised in Table 4. It is clear that both the magnitude and the sign of the reponse of each hydrometeor class to CDNC differs strongly not just between microphysics schemes, but also between cases. (Note that Table 4 shows relative amounts, and that the absolute difference in response to CDNC between each of the schemes and cases can also vary significantly). Whilst it is not surprising that the different cases of convection differ in their hydrometeor development and in their response to polluted conditions, it is worth noting the magnitude and variation of the difference in response. A body of literature uses idealised model configurations to investigate storm-system response to aerosol loading (e.g. Seifert and Beheng, 2006b; Khain and Lynn, 2009; Lebo and Seinfeld, 2011; Morrison, 2012) and to compare microphysics schemes (e.g. Lebo and Seinfeld, 2011). Our results highlight that the storm-system response in such a model configuration may not be representative of the response over larger spatiotemporal scales, supporting similar findings of larger-scale feedbacks and lifecycle-dependent responses in idealised (Morrison and Grabowski, 2011; Lee, 2012) and real-data (van den Heever et al., 2006) studies of aerosol–convection interations.

We note that the vertical resolution used in this study is relatively coarse, and that a horizontal grid length of 4 km is at the limit of what may be considered as 'convection-permitting' (Bryan et al., 2003). However, we use this grid spacing for consistency with a previous study, where 10 km and 4 km grid lengths were shown to be sufficient to reproduce storm characteristics and aerosol–convection interactions in the Congo basin (Gryspeerdt et al., 2015). Previous studies have indicated sensitivity of convection to horizontal grid spacing (e.g. Bryan and Morrison, 2012; Potvin and Flora, 2015) and also that the sensitivity to grid length can vary with microphysics scheme (Morrison et al., 2015b), although idealised ensemble studies of response to

aerosol have shown that differences between polluted and pristine conditions were similar in simulations using horizontal grid lengths of 4 km, 2 km, and 0.5 km, respectively and were also relatively robust to domain size (Morrison and Grabowski, 2011).

An important factor in our setup is that we use the same values of prescribed CDNC in all of our cases. Whilst the literature also shows widely varying response to aerosol, especially between bin and bulk schemes (where even the sign of the response may differ), Kalina et al. (2014) showed in idealised supercell simulations using 15 CCN concentrations and 4 environmental soundings that changes in cold pool characteristics with CCN were nonmonotonic and dependent on the environmental conditions. Therefore our use of the same CDNC values in multiple types of convection helps to minimise uncertainty due to nonmonotonic behvaiour. However, considering the results of Kalina et al. (2014), we note that a caveat of the present study (and indeed of the majority of existing studies) is that the absolute values of the cloud system and precipitation response to aerosol identified here may only hold for the CDNC values used in our study.

We find that the autoconversion representation alone is sufficient to explain most of the differences between microphysics schemes in the shallow cumulus case both in terms of their representation of cloud and precipitation (consistent with Li et al., 2015) and also in terms of their response to CDNC. The dominant hydrometeor difference between the microphysics schemes in the RICO simulations occurs in the rain - a different result from both the Congo basin configuration (where the dominant difference occurs in the liquid cloud) and the idealised supercell configuration (where the dominant difference occurs in the graupel). We also find that autoconversion of cloud droplets to rain is the mechanism that prevents the formation (or persistence) of liquid-phase cloud in the south of the domain in the Congo basin simulations using the Morrison scheme. This is in agreement with the study of Kalina et al. (2014), who found in idealised supercell simulations using the Morrison bulk microphysics scheme with a variable shape parameter for the raindrop size distribution that autoconversion rates decreased under CCN loading. The importance of autoconversion representation was shown by Gilmore and Straka (2008), who showed that the rates predicted by the autoconversion formulae used in bulk schemes differ by orders of magnitude. Modelling studies and observations from RICO have found that warm rain formation can be explained by the observed aerosol distribution (Blyth et al., 2013). In the context of our findings, this suggests that an accurate description of the autoconversion process in warm-rain regimes is fundamental not only to a realistic representation of cloud and precipitation, but also to its response to varying aerosol concentrations.

We caution that care should be applied when using autoconversion schemes in different regimes for which they were originally developed, such as the use of the Khairoutdinov and Kogan (2000) scheme for deep convective cases. Although not the focus of the present study, those interested in testing and improving autoconversion schemes could do so by calculating the mean volume radius and thereby the height of first raindrop formation through knowledge of the 13 to 14 micron critical radius for raindrop production (Freud and Rosenfeld, 2012; Khain et al., 2013; Rosenfeld et al., 2014). Similarly, comparison of results from bulk models to those from bin models (e.g. Thompson et al., 2004; Igel and van den Heever, 2017c) can also be a valuable tool for testing schemes. Based on the limited set of cases in our study, we would not be justified in recommending

one of the autoconversion schemes over the other. Moreover, because there are so many competing processes besides auto-conversion, including a number of microphysical and dynamical processes, it could be misleading to claim that one scheme is better than the other just based on bulk comparison with observations from a few cases. For those interested in testing and evaluating the autoconversion schemes, we suggest that the best approach would be to perform off-line testing based on detailed in-situ observations and calculations, as was done by e.g. Wood (2005), who tested the Khairoutdinov and Kogan (2000) autoconversion scheme in such a manner.

Our results (which are shown to hold across multiple cloud types and types of simulation) have important implications not just for cloud-resolving simulations but also for the global modelling community. Most significant, perhaps, is then the radiative impact which could arise when such major differences occur in the ice phase. Our Congo simulations illustrate just how large this uncertainty may be, and our tests using a bin scheme show that this is not purely an artefact of the bulk microphysics schemes used. Further, that uncertainties due to choice of microphysics scheme dominate any aerosol response within a given scheme has implications for global modelling studies of aerosol indirect effects (e.g. Zhang et al., 2016; Ghan et al., 2016). Once again, this highlights the continuing need of our community for tight observational contraints on cloud and precipitation processes and their repsonse to aerosol, and for ongoing parameterisation development to allow these processes to be accurately represented in large-domain (or global), long-term simulations.

*Acknowledgements.* This work used the ARCHER UK National Supercomputing Service (http://www.archer.ac.uk). The research leading to these results has received funding from the European Research Council under the European Union's Seventh Framework Programme (FP7/2007–2013)/ERC grant agreement no. FP7- 280025 (ACCLAIM) and grant agreement no. FP7- 306284 (QUARERE). The Congo precipitation data used in this study were acquired as part of the Tropical Rainfall Measuring Mission (TRMM). The algorithms were developed by the TRMM Science Team. The data were processed by the TRMM Science Data and Information System (TSDIS) and the TRMM office; they are archived and distributed by the Goddard Distributed Active Archive Center. TRMM is an international project jointly sponsored by the Japan National Space Development Agency (NASDA) and the US National Aeronautics and Space Administration (NASA) Office of Earth Sciences. The Congo radiance data used in this study were acquired as part of the Geostationary Earth Radiation Budget Project. The CloudSat data was obtained from the CloudSat Data Processing Center. ERA-Interim data provided courtesy ECMWF. Thanks go to Zak Kipling and Laurent Labbouz (University of Oxford, UK) for helpful comments on this manuscript.

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

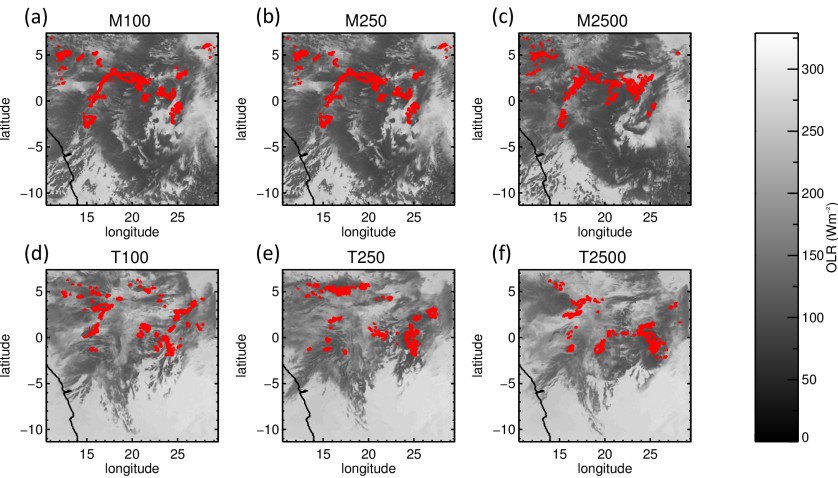

**Figure 1.** Congo case: Instantaneous outgoing longwave radiation (W m$^2$, greyscale) and 5 mm hr$^{-1}$ surface precipitation rate (red contour) at 0700 UTC, 7 August 2007 in the Congo basin configuration. (a), (b) and (c) show CONGO-MORR simulations, (d), (e) and (f) show CONGO-THOM simulations. Prescribed CDNC values of 100, 250 and 2500 cm$^{-3}$ are shown in (a, d), (b, e), and (c, f) respectively.

**Table 1.** List of model configurations

| Model settings | Congo | Supercell | RICO LES |
|---|---|---|---|
| horizontal grid length (km) | 4 | 4 | 0.1 |
| number of grid points (W–E and S–N) | 525 | 400 | 129 |
| number of vertical levels | 30 | 30 | 100 |
| model top | 5000 Pa | 20 km | 4 km |
| time step (s) | 12 | 12 | 1 |
| simulation length | 10 days | 2 hours | 24 hours |
| LW radiation scheme | RRTM | - | - |
| SW radiation scheme | Goddard | - | - |
| PBL scheme | YSU | - | - |

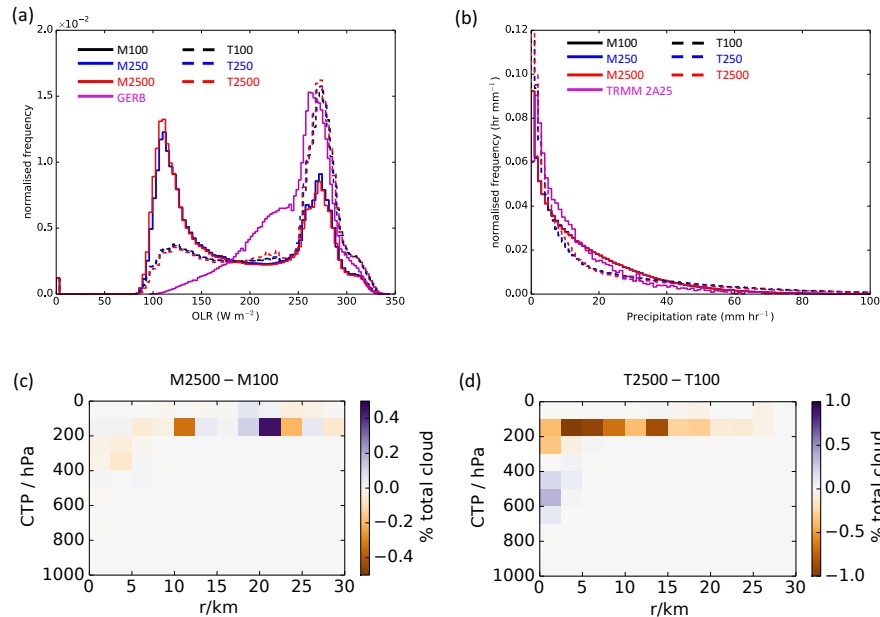

**Figure 2.** Congo case: (a) frequency distributions of OLR from the WRF simulations and observations from GERB over the period 01 to 10 August 2007 , (b) self-weighted precipitation rate distributions from the WRF simulations and observations from the ungridded TRMM 2A25 product, which has a similar spatial resolution to the 4 km model grid length, (c) difference in the joint distribution of cloud top pressure in updraughts (identified by masking points where the maximum vertical velocity exceeds 1 ms$^{-1}$, and then applying a connected-components labelling algorithm to identify unique updraught areas ) and horizontal radius of updraughts when CDNC is increased from 100 to 1000 cm$^{-3}$ using the Morrison microphysics scheme, (d) difference in the joint distribution of cloud top pressure in updraughts and horizontal radius of updraughts when CDNC is increased from 100 to 1000 cm$^{-3}$ using the Thompson microphysics scheme

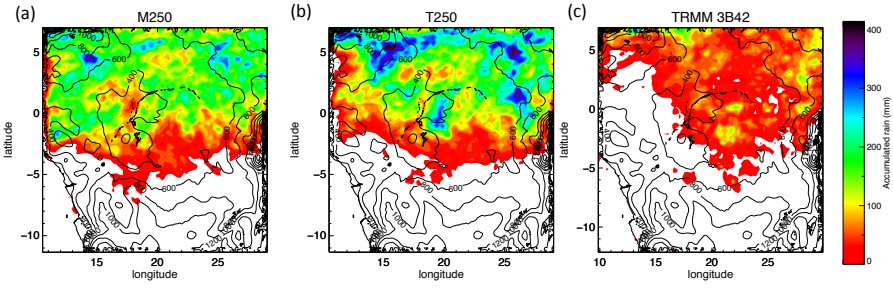

**Figure 3.** Congo case: accumulated surface precipitation (mm) from 01 to 10 August 2007 in the Congo basin, showing data from (a) CONGO-M250, (b) CONGO-T250 and (c) observations from the TRMM 3B42 gridded 3-hourly mean merged precipitation product. The simulation data shown in this Figure has been coarsened to the 0.25°spatial resolution of the TRMM product.

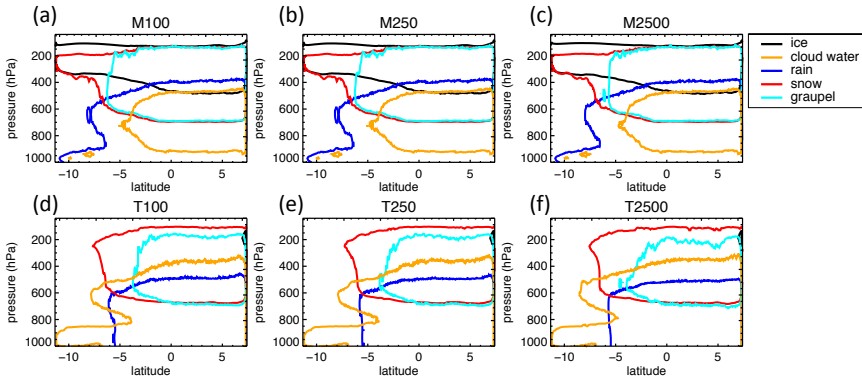

**Figure 4.** Congo case: Zonal mean vertical sections of hydrometeor classes (colour contours) from 01 to 10 August 2007. Hydrometeor mass mixing ratios are contoured at $10^{-6}$ kg kg$^{-1}$.

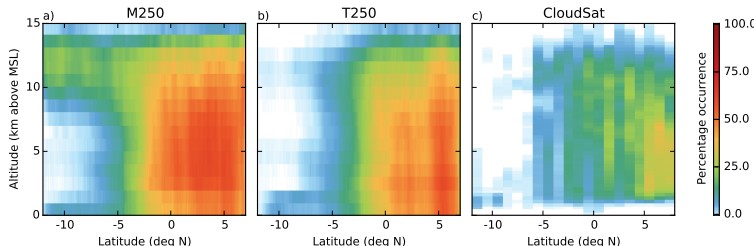

**Figure 5.** Congo case: 10-day histogram for the period 1–10 August 2007 of model reflectivities derived from hydrometeor fields passed through the Quickbeam radar simulator (Haynes et al., 2007), thresholded at values greater than -20 dBZ for (a) CONGO-M250, (b) CONGO-T250, and (c) the CloudSat 2B-GEOPROF product. In (a) and (b) the models have been sampled at the times of the nearest CloudSat overpasses.

**Table 2.** List of microphysics configurations tested, and the abbreviations used for each run

| Prescribed CDNC | Congo MORR | Congo THOM | Supercell MORR | Supercell THOM | RICO MORR | RICO THOM |
|---|---|---|---|---|---|---|
| 100 cm$^{-3}$ | CONGO-M100 | CONGO-T100 | SUPER-M100 | SUPER-T100 | RICO-M100 | RICO-T100 |
| 250 cm$^{-3}$ | CONGO-M250 | CONGO-T250 | SUPER-M250 | SUPER-T250 | RICO-M250 | RICO-T250 |
| 2500 cm$^{-3}$ | CONGO-M2500 | CONGO-T2500 | SUPER-M2500 | SUPER-T2500 | RICO-M2500 | RICO-T2500 |

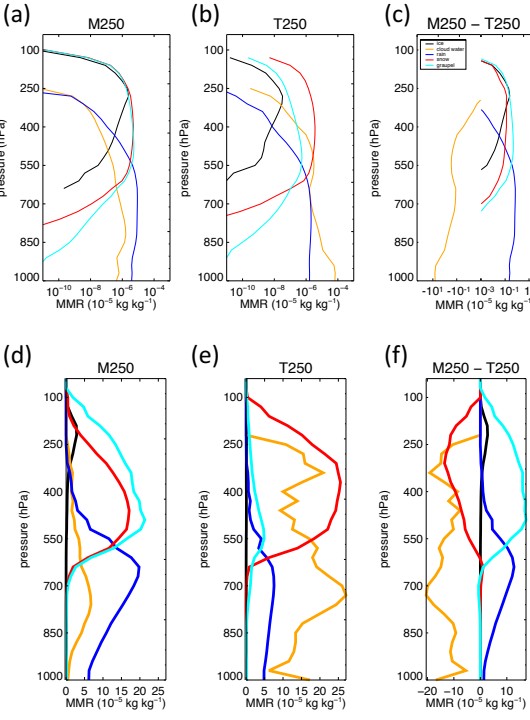

**Figure 6.** Congo case: Mean vertical profiles of hydrometeor mass mixing ratios (MMR) averaged over the period 1–10 August 2007. (a) CONGO-M250 cloudy column domain-mean, (b) CONGO-T250 domain-mean, (c) the difference in the domain-mean hydrometeor mixing ratio profiles (CONGO-M250 minus CONGO-T250), (d) CONGO-M250 mean over condensed points only, (e) CONGO-T250 mean over condensed points only for each hydrometeor class, and (f) the difference in the condensate-mean hydrometeor mixing ratio profiles (CONGO-M250 minus CONGO-T250). Note the logarithmic horizontal axis used in (a), (b) and (c) due to the total difference between the hydrometeor classes simulated by the two schemes spanning several orders of magnitude.

**Table 3.** Maximum relative difference of domain-mean hydrometeor mass mixing ratio profiles for the MORR and THOM schemes. The relative change in the hydrometeor mass mixing ratios are computed in each case for M250 minus T250.

| difference | CONGO | SUPERCELL | RICO |
|---|---|---|---|
| liquid cloud mass | -10900 % | -58.3 % | +17.0 % |
| ice mass | +98.7 % | +96.9 % | N/A |
| rain mass | +82.2 % | -138 % | -3830 % |
| snow mass | +40.8 % | -99.8 % | N/A |
| graupel mass | +91.6 % | +72.7 % | N/A |

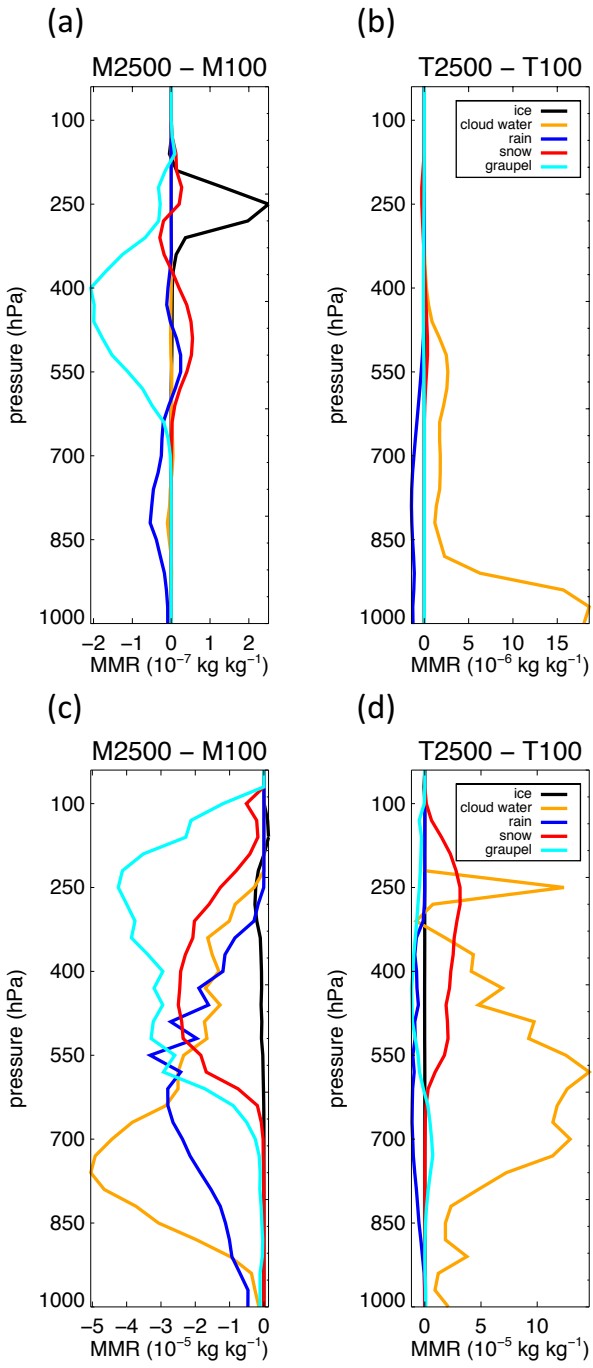

**Figure 7.** Congo case: Difference in the mean hydrometeor mixing ratio profiles under polluted and pristine conditions averaged over the period 1–10 August 2007. (a) CONGO-M2500 cloudy column domain-mean minus CONGO-M100 domain-mean, (b) CONGO-T2500 domain-mean minus CONGO-T100 domain-mean, (c) CONGO-M2500 mean over all condensed points of each hydrometeor class minus CONGO-M100 mean over all condensed points, and (d) ONGO-T2500 mean over all condensed points minus CONGO-T100 mean over all condensed points.

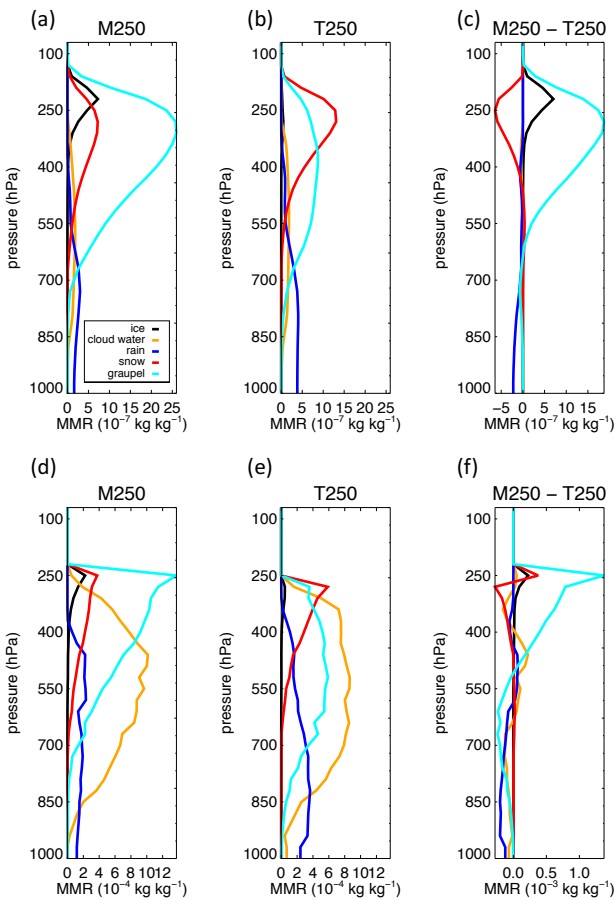

**Figure 8.** Idealised supercell: Mean vertical profiles of hydrometeor mass mixing ratios (MMR), as Figure 6, averaged over the 2 h of the supercell simulation. (a) SUPER-M250 domain-mean, (b) SUPER-T250 cloudy column domain-mean, (c) SUPER-M250 domain-mean minus SUPER-T250 domain-mean, (d) SUPER-M250 condensate-mean of each hydrometeor class, (e) SUPER-T250 condensate-mean, (c) SUPER-M250 condensate-mean minus SUPER-T250 condensate-mean.

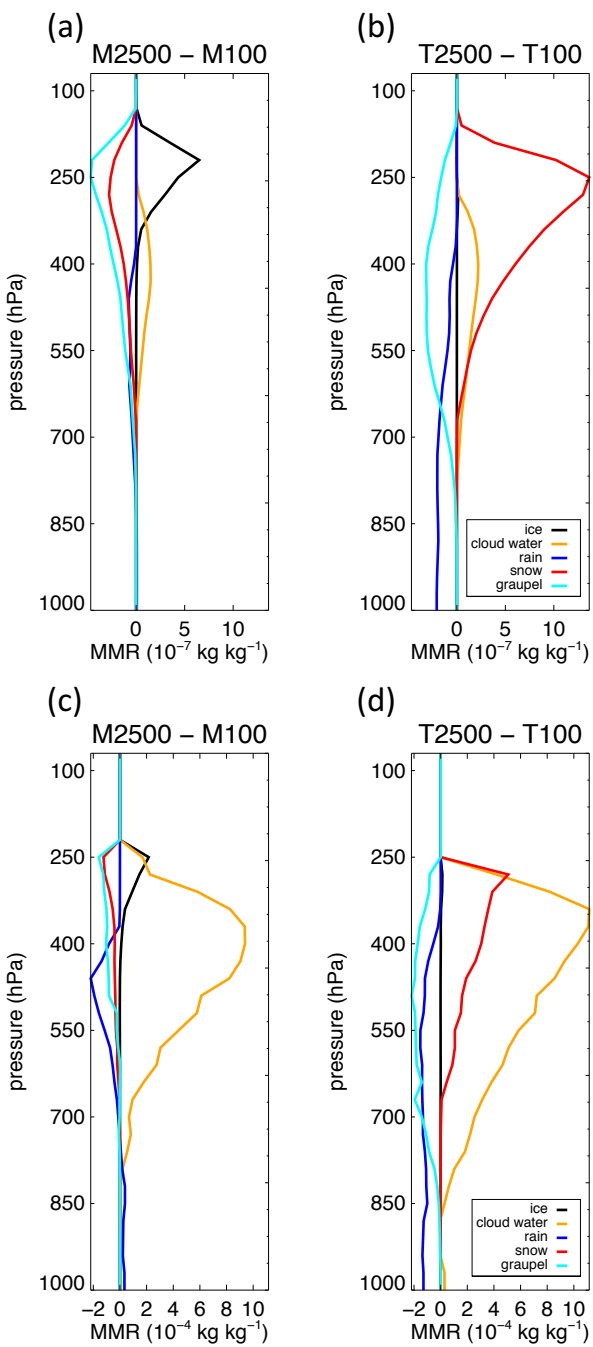

**Figure 9.** Idealised supercell: difference in the mean hydrometeor mixing ratio profiles under polluted and pristine conditions, as Figure 7, averaged over the 2 h of the supercell simulation. (a) SUPER-M2500 cloudy column domain-mean minus SUPER-M100 domain-mean, (b) SUPER-T2500 domain-mean minus SUPER-T100 domain-mean, (c) SUPER-M2500 condensate-mean of each hydrometeor class minus SUPER-M100 condensate-mean, and (d) SUPER-T2500 condensate-mean minus SUPER-T100 condensate-mean.

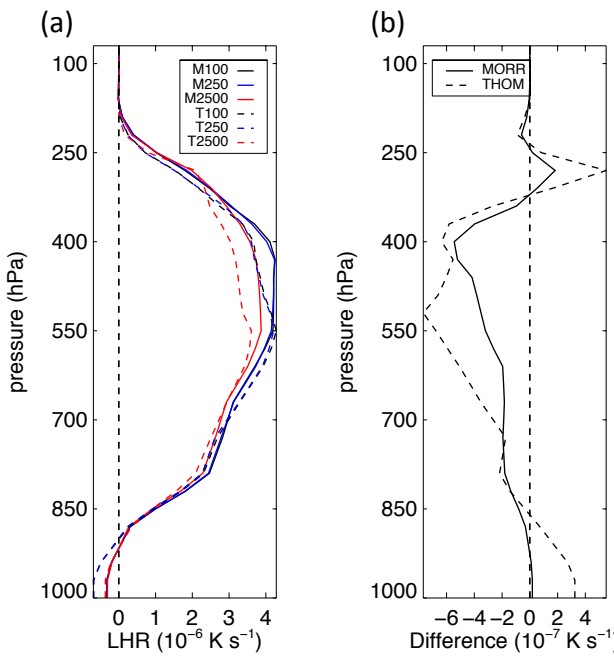

**Figure 10.** Idealised supercell: (a) Vertical profiles of domain-mean total latent heating rate (LHR) over the 2 h of the supercell simulation for SUPER-MORR and SUPER-THOM for CDNC values of 100, 250 and 2500 $cm^{-3}$. (b) Difference in the total latent heating contributions over the 2 h of the supercell simulation for SUPER-M2500 minus SUPER-M100 and SUPER-T2500 minus SUPER-T100.

**Table 4.** Maximum relative difference of response of model configurations to polluted conditions. The relative change in the domain-mean rehydrometeor mass mixing ratios are computed in each case for CDNC values of 2500 $cm^{-3}$ minus 100 $cm^{-3}$.

| difference | CONGO-MORR | CONGO-THOM | SUPER-MORR | SUPER-THOM | RICO-MORR | RICO-THOM |
|---|---|---|---|---|---|---|
| liquid cloud mass | -0.59 % | +32.2 % | +146 % | +169 % | -5.21 % | +44.0 % |
| ice mass | +12.5 % | -5.61 % | +116 % | +29.7 % | N/A | N/A |
| rain mass | -0.67 % | -62.6 % | -93.7 % | -51.6 % | -100 % | -100 % |
| snow mass | +1.37 % | +13.8 % | -33.5 % | +109 % | N/A | N/A |
| graupel mass | -4.60 % | -29.9 % | -19.1 % | -36.7 % | N/A | N/A |

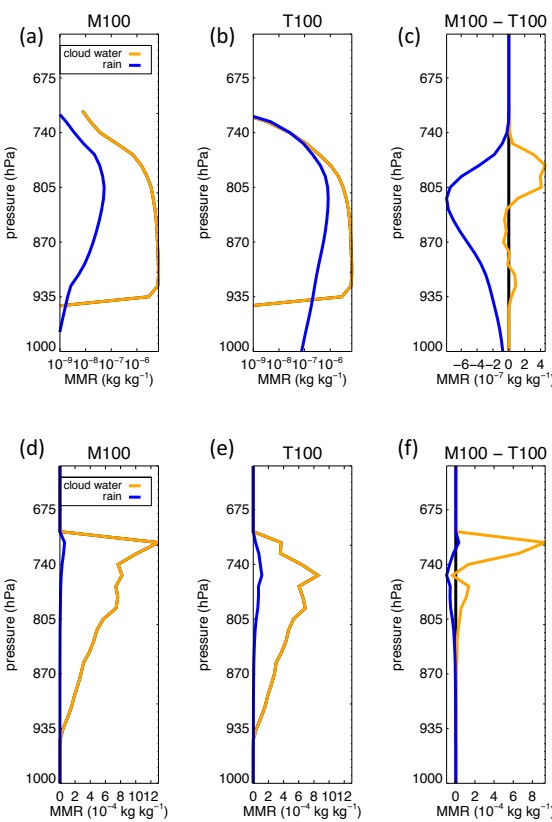

**Figure 11.** RICO case: Mean vertical profiles of hydrometeor mass mixing ratios (MMR), as Figure 6, averaged over the 24 h of the RICO simulation. (a) RICO-M100 cloudy column domain-mean, (b) RICO-T100 domain-mean, (c) RICO-M100 domain-mean minus RICO-T100 domain-mean, (d) RICO-M100 condensate-mean over each hydrometeor class, (e) RICO-T100 condensate-mean, (f) RICO-M100 condensate-mean minus RICO-T100 condensate-mean. Note that because the rain amounts are very small, especially in M100, (a) and (b) are shown with a logarithmic horizontal axis.

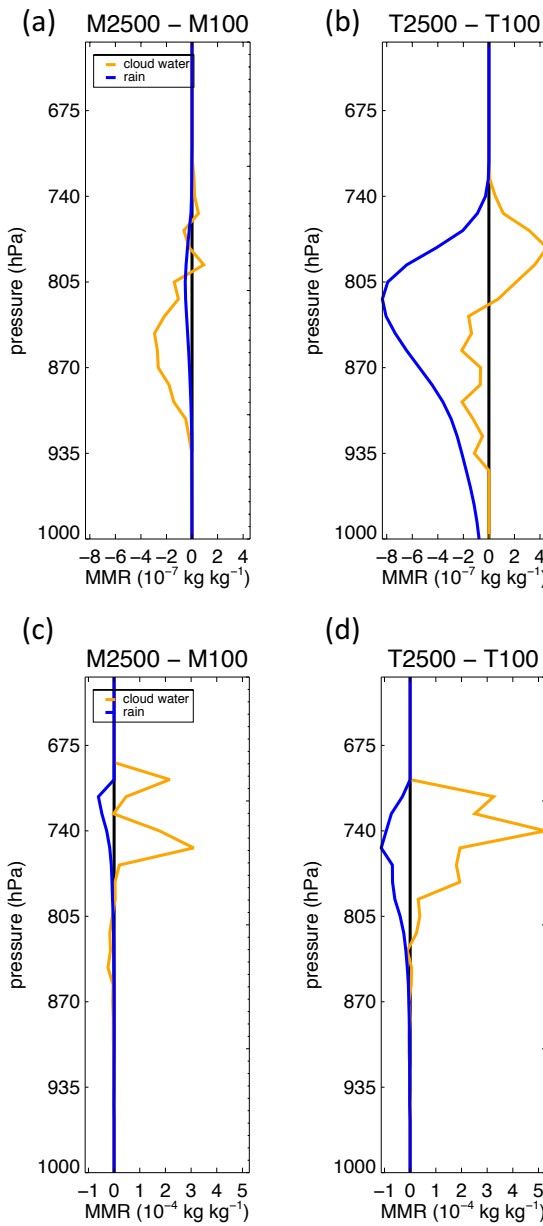

**Figure 12.** RICO case: difference in the mean hydrometeor mixing ratio profiles under polluted and pristine conditions in cloudy columns, as Figure 7, averaged over the 24 h of the RICO simulation. (a) RICO-M2500 cloudy column domain-mean minus RICO-M100 domain-mean, (b) RICO-T2500 domain-mean minus RICO-T100 domain-mean, (c) RICO-M2500 condensate-mean over each hydrometeor class minus RICO-M100 condensate-mean, (d) RICO-T2500 condensate-mean minus RICO-T100 condensate-mean

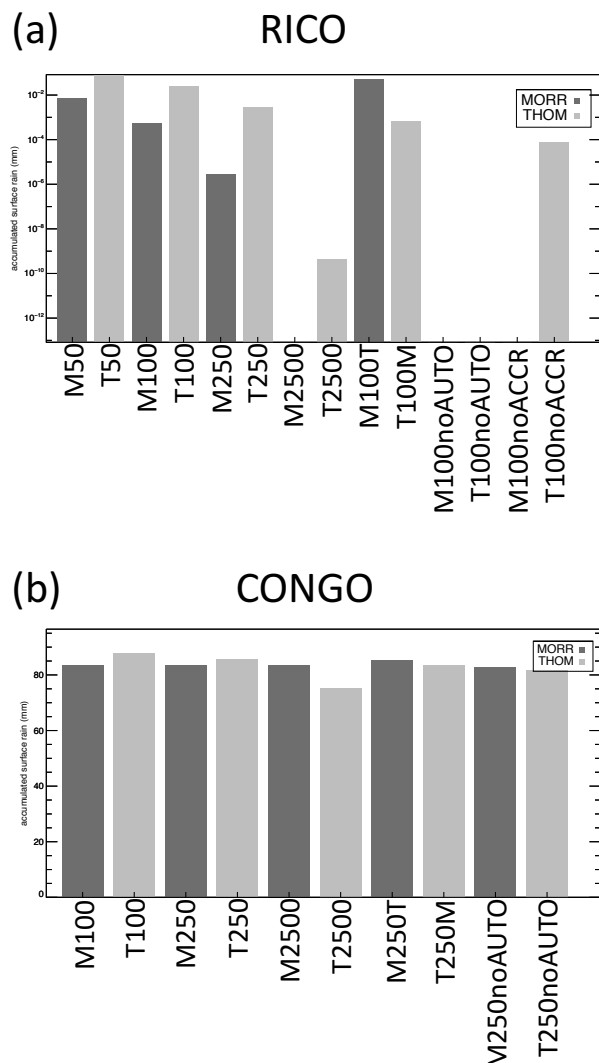

**Figure 13.** Total accumulated surface rain (mm) for each of the microphysics simulations, including a series of sensitivity simulations, for (a) RICO case, total after 24 h of simulation, (b) Congo case, total over the period 1–10 August 2007. Note that because the magnitude of the rain response to CDNC differs so strongly between the configurations in the RICO case, a logarithmic vertical axis is used in (a). The horizontal dashed line in (b) indicates the total precipitation from the TRMM 2A25 product over the same period.

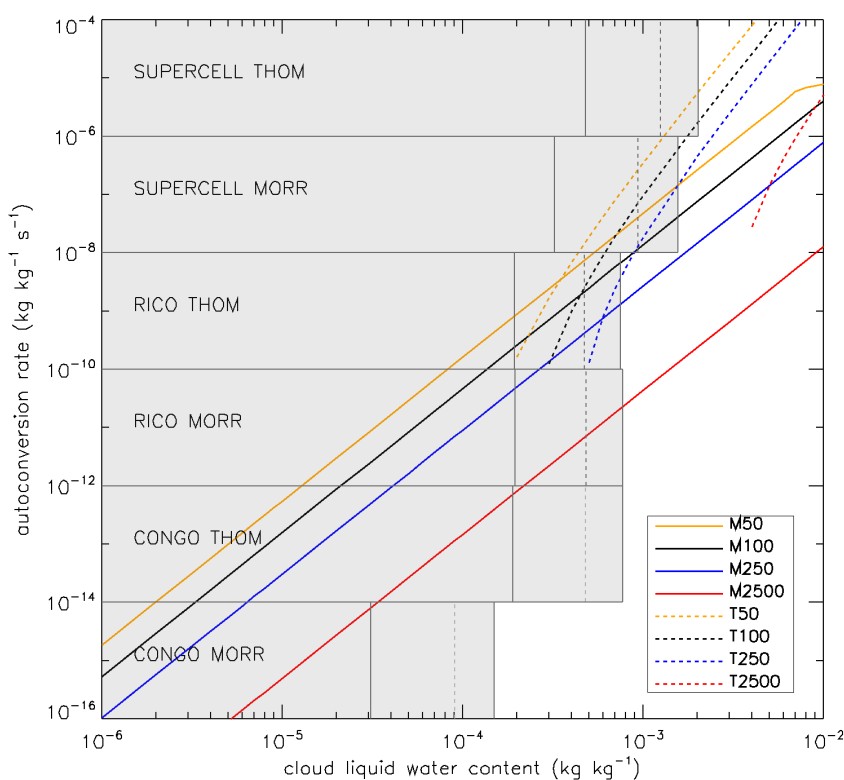

**Figure 14.** Autoconversion rate as a function of cloud water content for the MORR and THOM microphysics schemes (solid and dashed lines respectively) for super-pristine, pristine, moderately polluted and polluted conditions. Also shown are labelled grey bars showing the mean (solid vertical grey line) and 1 and 2 standard deviations (dashed vertical grey line and end of bar, respectively) cloud water content averaged over all prescribed CDNC configurations for each case (note that the variability in mean cloud water content with CDNC is significantly less than the variability due to microphysics scheme).

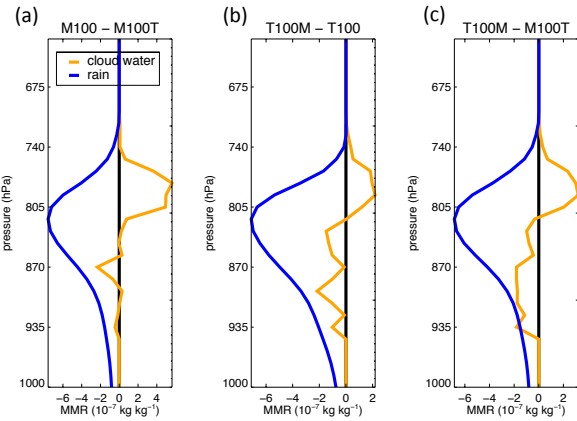

**Figure 15.** RICO case: Difference in the cloudy column domain-mean vertical profiles of hydrometeor mass mixing ratios (MMR) between MORR and THOM, as Figure 11c, averaged over the 24 h of the RICO simulation for the configurations with the autoconversion treatment swapped between the microphysics schemes: (a) M100 minus M100T, (b) T100M minus T100, (c) T100M minus M100T

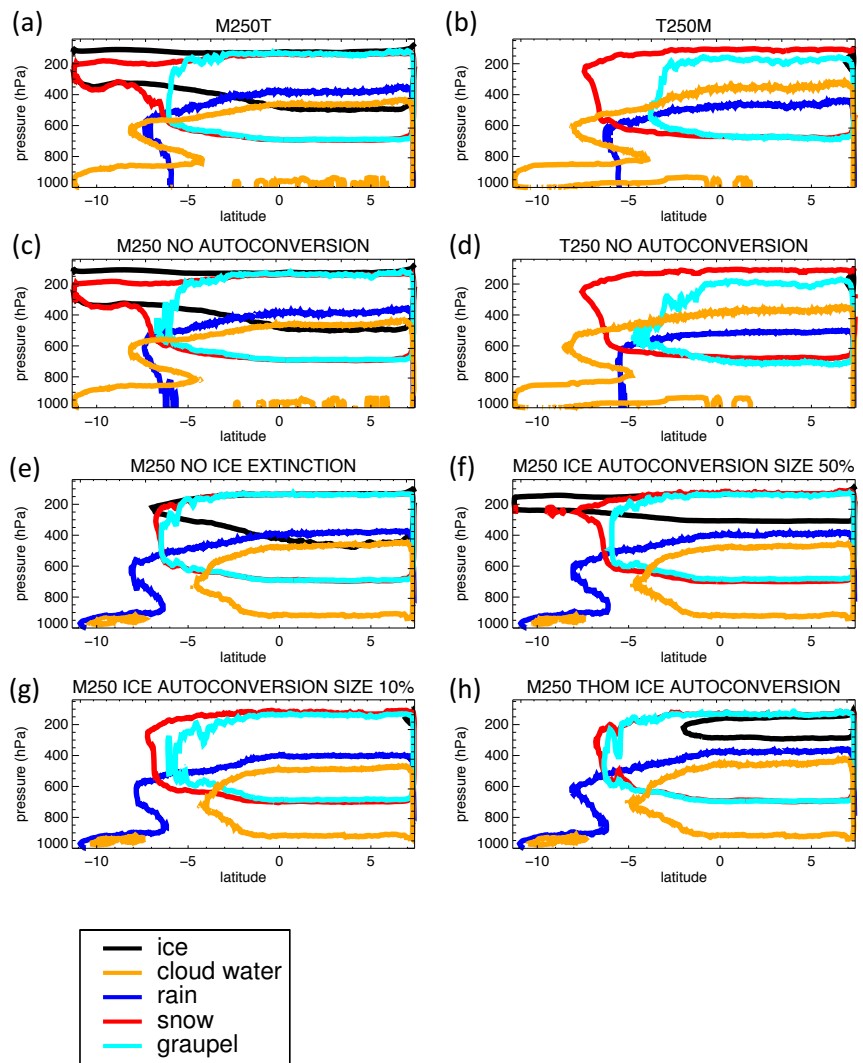

**Figure 16.** Congo case: Zonal mean vertical sections of hydrometeor classes (colour contours) from 01 to 10 August 2007, as Figure 4, but for the configurations with the autoconversion treatment swapped between the microphysics schemes, (a) CONGO-M250T and (b) CONGO-T250M, and for the configurations with (c) CONGO-M250 with autoconversion turned off, (d) CONGO-T250 with autoconversion turned off, (e) CONGO-M250 with the ice extinction coefficient set to zero in the longwave and shortwave radiation schemes, (f) CONGO-M250 with the threshold size parameter for conversion of ice to snow reduced to 50% of its default value, (g) CONGO-M250 with the threshold size parameter for conversion of ice to snow reduced to 10% of its default value, (h) CONGO-M250 with the autoconversion of ice to snow replaced by that used in the Thompson microphysics scheme. Hydrometeor mass mixing ratios are contoured at $10^{-6}$ kg kg$^{-1}$.