# Peer review of "Uncertainty from choice of microphysics scheme in convection-permitting models significantly exceeds aerosol effects"

_Atmospheric Chemistry and Physics, 2016_

## Referee Comment (RC1) · Anonymous Referee #1 · 6 Oct 2016

Review of the paper "Can models robustly represent aerosol–convection interactions if their cloud microphysics is uncertain?", authored by B. White, E. Gryspeerdt, P. Stier, H. Morrison, and G. Thompson.

The authors have drawn a pretty grim picture of the situation with the description of microphysics in cloud resolving models using bulk parameterization schemes. Two known bulk parameterization schemes referred to as "MORR" (Morrison and Milbrandt (2011) and "THOM" (Thompson et al. (2004, 2008)) were tested by simulation of 3 case studies with different type of convection: from shallow convection to a supercell storm. The simulations were performed at a priory given droplet concentrations of 100 cm-3, 250 cm-3 and 2500 cm-3. A dramatic difference between parameters of simulated

clouds (cloudiness, simulated precipitation, etc.) and observations is demonstrated. A huge difference in cloud microphysical structure simulated by these two schemes is reported. Both bulk schemes turned out to be insensitive to droplet concentration. So, the difference in results are related to the differences in the bulk-parameterization schemes.

My particular comments and remarks are the following.

1. The finding about high diversity of cloud microstructure and precipitation produced by different bulk schemes is not new. Any intercomparison study shows such diversity. The insensitivity of most bulk schemes to aerosols (and to droplet concentration) also well known. For instance, application of different bulk -parameterization schemes to simulation of hurricane Irene (2011), including "MORR" and "THOM", led to a TC with maximum wind varying from 30 m/s to 70 m/s (Khain et al., 2015; Khain et al. 2016-Atmospheric Research 167, 129–145). In these studies insensitivity of bulk schemes to aerosols was also demonstrated and an possible explanation of such insensitivity is presented .

2. The finding that description of the autoconversion is of crucial importance for correct simulation of cloud microphysics is also not new. For instance, Gilmore and Straka (2008) showed that the most formulae for autoconversion used in bulk schemes are applicable to the initial stage of the first raindrop formation only, and that the rates predicted by these formulae differ by orders of magnitude. There are many other studies showed dramatic sensitivity of cloud microstructure to the scheme of autoconversion.

So, the new information in the paper is that these conclusions are confirmed in investigation of these two particular bulk schemes.

3. The authors illustrate vertical profiles of mass contents of different hydrometeors averaged over the entire computational area. As a result, all information concerning the microphysical structure of clouds simulated by different schemes turns out to be lost, at least for specialists in cloud microphysics. It is necessary to present vertical profiles of

maximum values of the mass contents. The profiles of cloud averaged values would be also useful. These figures should be accompanied by corresponding comments and analysis.

4. The lack of physical interpretation of results is another drawback of the study. For instance, Fig. 6 and Fig. 7 show that in simulations CONGO - T250, T-2500 maximum cloud water content (i.e. small cloud droplets) is located at the surface. One gets the impression that the entire boundary layer is filled with tiny droplets (but not with raindrops). What can be physical mechanisms leading to this very strange effect?

The comment concerning the lack of interpretation is related to most figures.

5. The section of Conclusions is weak. The authors stress that their key findings are a) "A key finding is that the simulated hydrometeor classes differ significantly between microphysics schemes" b) "Another key finding is that the difference between the hydrometeor classes simulated by each microphysics scheme varies between cases of convection." c) Another key finding is that the cloud morphological difference and the difference in the hydrometeors between different schemes is significantly larger than that due to CDNC perturbations.

As it was said above, all these findings are not new.

I would recommend to rewrite conclusions by adding more detailed analysis of results and recommendations of the ways to improve the schemes.

I recommend major revision before the decision on publication will be discussed.

Please also note the supplement to this comment:
http://www.atmos-chem-phys-discuss.net/acp-2016-760/acp-2016-760-RC1-supplement.pdf

---

## Referee Comment (RC2) · Anonymous Referee #2 · 23 Oct 2016

Review of the ACP manuscript "Can models robustly represent aerosol–convection interactions if their cloud microphysics is uncertain?" by White et al.

The authors used the WRF model coupled with two double-moment bulk microphysics scheme to perform cloud system-resolving simulations of convection in the Congo basin, an idealized supercell, and a case of shallow cumulus convection, and tested the sensitivity of the simulated hydrometeors and precipitation to the microphysics scheme and to CDNC perturbations. The authors showed the simulations are sensitive to microphysics parameterizations much more than CDNC, which has been showed in previous studies but this study highlighted this point to imply aerosol effects are the secondary compared with the uncertainty in cloud parameterization. They further examined the shallow cumulus convection case and found that representation of autoconversion is the dominant factor that drives differences in rain production. The paper is generally written clearly but there are confusing sentences. They are some perspectives that can not be well justified physically such as saying the aerosol-cloud interactions in their study represent the upper limit.

The major problem is that their main point is based on an assumption that those two-moment schemes well represent the aerosol-cloud interactions, which is not the case based on the many past studies and a recent review paper by Khain et al 2015. Two-moment schemes have significant limitations in aerosol-cloud interaction process parameterizations such as nucleation, diffusional growth, and sedimentation, etc (detailed Khain et al. 2015). The paper did not really address the question "Can models robustly represent aerosol–convection interactions if their cloud microphysics is uncertain", so that the title needs to be changed. More literature survey is needed, especially about those studies comparing different microphysics schemes and their responses to CCN or CDNC. Those studies need to be discussed in the introduction and the relevant places in the paper, especially about some important points on the problems with the parameterizations of some specific microphysics processes in the bulk schemes. Therefore, the paper needs major revisions to be accepted as a publication in ACP.

Specific comments:
1. The title needs to be changed. It is relevant but the authors did not conduct an unique study to really address this question. The two-moment schemes can not robustly represent aerosol–convection interactions due to the limitations in representing the most relevant processes as detailed in Khain et al. 2015. If a bin scheme is used, you might end up with similar magnitudes of aerosol indirect effects as the differences among different microphysics schemes. In addition, for specific case simulations, one can not really reveal how it is impacted by aerosols by conducting simulations with realistic aerosols/chemistry configuration. Any aerosol properties and spatial distribution change could change cloud and precipitation.

2. P1, L21-22: aerosol can affect through aerosol radiative effects as well.
3. P1, L24: Albrecht, 1989 actually showed the suppression of precipitation but for warm clouds. So, the sentence is not accurate.
4. P2, L29: "Until recently" should be deleted.
5. P4, L19-25, these sentences could be misleading. First, it is not clear what the authors

mean by saying "the response of different microphysics schemes to perturbations in prescribed cloud parameters". Second, some past studies showed qualitatively different aerosol impact for different cloud types, with a purpose of illustrating a point that aerosol impacts depend on cloud type and dynamical and thermodynamic conditions of each case. So it is misleading to describe a study without giving information about cloud types or specific dynamical and thermodynamic environment. For example, the description about Fan et al. 2012 about aerosol reducing precipitation is not correct. The study did two different cloud cases over the eastern China – one deep convective cloud case with warm cloud base and weak wind shear and the other a winter stratiform cloud case, with aerosol increased precipitation for the former but reduced precipitation for the latter.

6. P5, L3: cloud and precipitation responses > cloud and precipitation responses to perturbation of CDNC

7. P5, L25: Khain and Lynn 2009 is not s study with specified CDNC. CDNC is prognostic in the bin model they used.

8. P6, first paragraph, RRTM and Goddard schemes only talk hydrometeor mass from microphysics calculation (Goddard shortwave scheme takes prognostic CDNC as well). The microphysics-radiation coupling does not account for particle size changes, which means some aerosol effects are missing in those studies.

9. Figure 3, the color scheme needs to be changed. The color difference is too little even between 0 and 100 mm.

10. Figure 5, why compare with the climatology data? This is just a 10-day run, how should we expect it represent the climatology?

11. P. 9 L5-7, this sentence is confusing. Need to be clarified.

12. P. 9, L19: domain averaged cloud water in T250 is 140 times larger than M250. Something could be wrong here. Can you plot the cloudy-point average for cloud water mass total hydrometeor mass to check if they make sense? What is the maximum cloud water mass in T250 and M250, respectively?

13. Figure 6, this figure need to be replotted to show differences of other hydrometeor mass clearly. Right now, only cloud water differences can be seen clearly. I would use different panels for different hydrometeors.

14. P9, last paragraph and Figure 7, the huge increase of cloud water with the increase of CDNC with the Thompson scheme seems not reasonable. How about the change of precipitation?

15. P. 10, L19-21: The sentence "we also see that the simulated hydrometeor classes differ between cases: the difference in the simulated hydrometeor classes in the idealised supercell configuration is different from the difference in the real-data Congo basin configuration" is not necessary. This is what it should be since they are different convective cloud types.

16. P11, first paragraph, the sentences in L5-6 and in L10-11 are repeated.

17. P11, first paragraph, the main point here should be about more significant aerosol impact on hydrometeor mass on the supercell case compared with the Congo case, not the different responses of hydrometeors between the CONGO case and the superell case, since they should be expected for completely different cases. Many past studies have showed that aerosol impacts depend on dynamics and thermodynamics of convective clouds (e.g., Khain 2009; Fan et al. 2009).

18. P11, last paragraph: the lack of appropriate sensitivity of bulk schemes to aerosols is mainly due to the limitation of bulk scheme parameterization in nucleation, diffusional growth, and sedimentation, etc, as detailed in Khain et al. 2015. The invigoration of updrafts can not be simulated since the saturation adjustment approach for diffusional growth of droplets limit such effects. Those aspects should be considered and discussed when interpreting the results on aerosol indirect effects here. Past studies showing the limitation of bulk scheme parameterizations in representing aerosol-cloud interactions need to be surveyed and discussed.
19. P12, L5-7: Again physically it is supposed to be that for different types of convective cases. Hydrometeor differs and hydrometeor responses to CDNC are different as well.
20. P13, L23-25, reword the sentence. Not sure what you really want to say.
21. P15, first paragraph, the authors showed the autoconversion process is not the significant process contributing to the large cloud mass at low –levels. Then the question is what process mainly contributes to it?
22. P15, L19-20, again, the limitation of two-moment bulk schemes in representing aerosol impacts on microphysics processes should be discussed.
23. P17, L21-23, this sentence appears in a few places throughout the study, but the point can not be well justified even for aerosol indirect effects. First, you not know what the reality of aerosol look like in composition and spatial variability, many studies showed that aerosol spatial distribution could significant change storm location such as urban aerosols impact significantly on the precipitation in the downwind area of cities through aerosol indirect effects. Second, since two-moment bulk schemes even can not represent aerosol-cloud interaction processes physically, then how do you justify the aerosol impact here represent the upper limit?
24. P18, L29, this is definitely not the first study to consider two and more cloud cases. A thorough literature search would give you those past studies.
25. P18, L30-31, again, it has been a basic understanding that hydrometeors and their responses to CCN or CDNC vary with different cloud types and convective cases.

---

## Author Comment (AC1) · 24 Dec 2016

Review of the paper "Can models robustly represent aerosol–convection interactions if their cloud microphysics is uncertain?", authored by B. White, E. Gryspeerdt, P. Stier, H. Morrison, and G. Thompson.

The authors have drawn a pretty grim picture of the situation with the description of microphysics in cloud resolving models using bulk parameterization schemes. Two known bulk parameterization schemes referred to as "MORR" (Morrison and Milbrandt (2011) and "THOM" (Thompson et al. (2004, 2008)) were tested by simulation of 3 case studies with different type of convection: from shallow convection to a supercell storm. The simulations were performed at a priory given droplet concentrations of 100 cm-3, 250 cm-3 and 2500 cm-3. A dramatic difference between parameters of simulated clouds (cloudiness, simulated precipitation, etc.) and observations is demonstrated. A huge difference in cloud microphysical structure simulated by these two schemes is reported. Both bulk schemes turned out to be insensitive to droplet concentration. So, the difference in results are related to the differences in the bulk-parameterization schemes.

My particular comments and remarks are the following.

1. The finding about high diversity of cloud microstructure and precipitation produced by different bulk schemes is not new. Any intercomparison study shows such diversity. The insensitivity of most bulk schemes to aerosols (and to droplet concentration) also well known. For instance, application of different bulk -parameterization schemes to simulation of hurricane Irene (2011), including "MORR" and "THOM", led to a TC with maximum wind varying from 30 m/s to 70 m/s (Khain et al., 2015; Khain et al. 2016-Atmospheric Research 167, 129–145). In these studies insensitivity of bulk schemes to aerosols was also demonstrated and an possible explanation of such insensitivity is presented.

We agree with Reviewer 1 that it has been shown that different microphysics schemes produce a high diversity of cloud development and precipitation. We have significantly revised our introduction to show this more explicitly, and in doing so have included the references and possible explanations suggested by the Reviewer. This is in P1 L20 through P8 L4 of our revised manuscript.

We certainly do not intend to claim this in itself as a new result. However, we confirm this in a single modelling framework, for three different cloud and environment types, in three different simulation types (one performed with meteorological data, one idealised supercell using open boundaries, one warm-phase cumulus case with periodic boundaries). This is necessary to show before presenting our main results: (a) that the largest source of uncertainty (or variability) between simulations is in the choice of microphysics scheme, and that any aerosol effect is secondary, and (b) that the response of the hydrometeors to CDNC perturbations differs strongly not just between microphysics schemes but also that the inter-scheme variability differs between cases of convection

Although they do not always respond to aerosol as strongly as bin schemes, bulk schemes have been shown to be sensitive to aerosol, e.g. Morrison & Grabowski 2011, who found an ice-phase response to aerosol. Indeed, a similar mechanism was later confirmed in bin-scheme simulations by Fan et al. 2013. Further, Kalina et al. 2014 found that autoconversion of cloud water to rain decreased under polluted conditions and, subsequently, near-surface rain and hail particles increased in size due to enhanced collection of cloud droplets. Although the sensitivity of the bulk schemes used in the current study to perturbations in cloud droplet numbers was secondary to the sensitivity in cloud and precipitation

development dependent on the choice of bulk scheme, we nevertheless feel this is a significant result to present in the context of multiple cloud types and types of simulation, given that bulk microphysics schemes remain in wide use.

Futher, in idealised supercell simulations using the WRF-SBM, we find the same order of magnitude of aerosol effect as in our bulk scheme simulations, and uncertainty due to microphysics scheme still dominates aerosol effects (Figure S4 in the Supplement).

We present in Figure S1 in the Supplement joint histograms for our Congo basin simulations of cloud top height in convective updraughts (columns identified with vertical velocities greater than 1 ms$^{-1}$) and the radius of the updraughts, as identified by running a connected-components labelling algorithm on the field of identified updraughts. Figure S1 shows that the most significant dynamical difference comes from choice of microphysics scheme: the Morrison scheme has a tendency towards higher frequencies of wider updraught radii with higher cloud tops than the Thompson scheme.

We note that the updraught dynamics in the Congo simulations respond very differently to aerosol. Figure R2 shows the difference between the joint histograms for the polluted and pristine cases for each microphysics scheme, respectively. While MORR shows little consistent aerosol response, THOM shows consistently lower frequencies of occurrence of all updraught radii and a reduced frequency of occurrence of the highest updraught tops under polluted conditions, with an increased frequency of occurrence of small updraught radii with lower cloud tops. Therefore, a consistent aerosol response is observed in THOM, resulting in smaller and lower convective updraughts (i.e. weakened convection under polluted conditions). Interestingly, both of these effects contradict the findings of Morrison & Grabowski 2011, who found an ice-phase response to aerosol in which cloud top heights and anvil ice mixing ratios increase under polluted conditions due to increased freezing of larger numbers of cloud droplets and subsequent higher ice particle concentrations with smaller sizes and reduced fall speeds. However, we note again that we consider different values of CDNC / CCN (and responses may be nonmonotonic, Kalina et al. 2014), and different case of convection (indeed, our 10-day Congo simulation covers many convective lifecycles). This combined with the findings already presented in our study, leads us to suggest that it is not certain that a consistent response in a different case of convection and with different CDNC values would be expected.

[Figure]

**Figure R2:** *Congo case: difference in joint histograms of updraught radius and cloud top pressure at the top of the cloudy updraughts between the polluted and pristine cases for MORR (left) and THOM (right).*

We have included Figure R2 in the revised paper and the relevant discussion Is on P11 L11-30.

2. The finding that description of the autoconversion is of crucial importance for correct simulation of cloud microphysics is also not new. For instance, Gilmore and Straka (2008) showed that the most formulae for autoconversion used in bulk schemes are applicable to the initial stage of the first raindrop formation only, and that the rates predicted by these formulae differ by orders of magnitude. There are many other studies showed dramatic sensitivity of cloud microstructure to the scheme of autoconversion. So, the new information in the paper is that these conclusions are confirmed in investigation of these two particular bulk schemes.

We thank Reviewer 1 for the Gilmore & Straka reference which helps to give context to our findings. We have included this reference in our Results (P18 L10) and Conclusions (P25 L24) section.

We do not claim our finding that cloud structure is sensitive to autoconversion representation to be a new result. Rather, we find that autoconversion is sufficient to explain some of the key differences between the clouds simulated by the two bulk microphysics schemes, namely those in the liquid phase. Further, we show that aerosol effects are dominated by the response of the autoconversion process for this case. Li et al. (2015, JAS) also showed that a slower autoconversion process along with a stronger accretion process explains the Morrison scheme's higher cloud fraction than the Thompson scheme for a similar rain mixing ratio.

This is in part a finding in itself that addresses Reviewer 1's comment 4. It is because we were trying to understand which processes were significant to the differences in cloud microstructure in the context of comparing these two schemes that we investigated the autoconversion representation. Indeed, we cannot say in the cases we consider which is the more 'correct' representation. We rather highlight that some of the differences we observe between the schemes, in some of the types of convection, can be almost entirely attributed to the representation of autoconversion, and that differences in other processes between the two schemes are secondary.

Further, our tests highlighted the difference between the MORR and THOM schemes in the interplay and relative importance of the autoconversion and accretion processes: the Thompson scheme can produce surface rain from autoconversion alone (although two orders of magnitude less than when rain can also accrete cloud water), showing that autoconversion acts almost like a 'seed' for rain production in this scheme, after which accretion takes over the rain production process. However, in the Morrison scheme, when autoconversion is allowed to occur but accretion of cloud water by rain is prevented, precipitation shuts down, thus indicating that both autoconversion and accretion are necessary processes for warm rain production in MORR.

This is discussed in our Results section.

3. The authors illustrate vertical profiles of mass contents of different hydrometeors averaged over the entire computational area. As a result, all information concerning the microphysical structure of clouds simulated by different schemes turns out to be lost, at least for specialists in cloud microphysics. It is necessary to present vertical profiles of maximum values of the mass contents. The profiles of cloud averaged values would be also useful. These figures should be accompanied by corresponding comments and analysis.

It is important to include the profiles averaged over the entire domain, because (at least for larger-scale impacts) this illustrates the difference in the bulk properties and accounts for differences in total cloud cover, which in turn has implications for radiative effects and

feedbacks. This is especially important in our Congo simulations, where differences in the ice-phase microphysics between the schemes lead to large differences in the ice cloud fraction, which will in turn have a significant radiative impact.

We agree with Reviewer 1 that it would also be useful to see the mean in-cloud properties, and changes thereof. We thus now include condensate-averaged profiles for each hydrometeor type in order to identify not only the bulk impact of changes in CDNC in each simulation but also the bulk properties of the cloud in each convective study. We note that in the idealised supercell case and the RICO LES case there is little difference between the domain-mean profiles and the hydrometeor class-mean profiles.

Comments and analysis of these new profiles have been included in our revised manuscript. The relevant Figures are Figures 6,7,8,9,11,12.

An example of the condensed-point average profiles is provided in our response to Reviewer 2's comment 12.

However, we do not agree that showing domain-maximum values is useful, because the maximum values of the mass contents represent just one point in the entire domain and can lead to improper conclusions, especially in our Congo simulation where multiple cloud types exist in the same domain but do not necessarily interact.

4. The lack of physical interpretation of results is another drawback of the study. For instance, Fig. 6 and Fig. 7 show that in simulations CONGO - T250, T-2500 maximum cloud water content (i.e. small cloud droplets) is located at the surface. One gets the impression that the entire boundary layer is filled with tiny droplets (but not with raindrops). What can be physical mechanisms leading to this very strange effect? The comment concerning the lack of interpretation is related to most figures.
The low cloud extending down to near-surface regions is indeed a strange effect. However, it is produced consistently in all simulations (including the simulation performed with WRF SBM microphysics, see Figures S2 and S3 in the Supplement now included with this response). We note also that in our WRF-SBM simulation (Figures S1 and S2 in the Supplement) the warm cloud mass has significantly greater domain-mean values than the M250 case. We believe this difference is because, as we have shown (our Figure 16), the cloud mass in the M250 case is removed through autoconversion to rain, whereas this does not occur in the T250 case (or indeed the SBM). When the THOM autoconversion is used in the equivalent MORR simulations (or when autoconversion is turned off completely in MORR), the cloud droplets persist as in THOM. Although we have not tested the equivalent process in the SBM simulation presented in this response, we suggest that lack of autoconversion (or, stronger autoconversion than THOM but weaker than that in MORR) is responsible for the persistence of the cloud droplets in THOM. That all schemes produce the warm cloud mass, but some configurations rain it out if the autoconversion rate is fast enough, leads us to suggest that it is not the Thompson scheme per se which is responsible for producing the low-level cloud mass, but rather the meteorological conditions present in which these simulations are performed. We provide in our response to Reviewer 2's comment 21 a figure (Fig. R4 in the response to Reviewer 2) figures and further discussion on the presence of this low warm cloud mass.

We have now included such comments on the presence of the warm cloud mass in our Results (P20 L10-16) section.

However, we disagree with Reviewer 2's comment that there is a general lack of interpretation. In the liquid cloud (warm cloud mass in the south of the Congo domain and the cloud in the RICO simulations) the differences can be attributed to the autoconversion, as we have already shown in our tests and describe in the discussion of our figures. In the deep convective cloud (where ice processes can occur), much of the difference between the schemes can be attributed to differences in the classification of frozen particles, which we have already discussed at length.

In addressing the Reviewer's comments on lack of interpretation, we have now performed further simulations which test the autoconversion of cloud ice to snow in the two schemes. These are presented and discussed in our revised manuscript and can be found from P20 L17 to P21 L2.

5. The section of Conclusions is weak. The authors stress that their key findings are a) "A key finding is that the simulated hydrometeor classes differ significantly between microphysics schemes" b) "Another key finding is that the difference between the hydrometeor classes simulated by each microphysics scheme varies between cases of convection." c) Another key finding is that the cloud morphological difference and the difference in the hydrometeors between different schemes is significantly larger than that due to CDNC perturbations.

As it was said above, all these findings are not new.
I would recommend to rewrite conclusions by adding more detailed analysis of results and recommendations of the ways to improve the schemes.
We present the new results that the way in which the schemes differ from each other between cases of convection is not systematic, and further that their response to aerosol also differs non-systematically.

By performing simulations within a single modelling framework we reduce much of the uncertainty in comparing results from different microphysics schemes noted by Khain 2015, such as orography, boundary layer parameterization, etc. We note that we have only performed real-data simulations in one region over one 10-day period in August 2007 and therefore would not expect our results to be exactly the same in another region or in another season, where orography, large-scale meteorology, and cloud type would be different from that present in our simulations. However, our main result is that cloud impacts from choice of microphysics scheme far exceed cloud impacts from CDNC perturbations in every case we consider, and therefore we would expect that even though the cloud development and response to CDNC would clearly be different in another regime, the uncertainty due to choice of microphysics scheme would still dominate.

We have rewritten our Conclusions, and put our results in the context of a wider body of previous work. These can be found in the revised manuscript from P21 L21 through P26 L4.

We stress that without observations it is difficult to suggest ways to improve the schemes. However, we have now performed extra tests to explain the processes which lead to the large differences in upper-level ice in the Congo simulations. This is discussed in our Results section from P20 L18 through P21 L2, along with suggestions as to how the schemes could be improved.

Further, we also find the same persistent upper-level ice in our WRF-SBM Congo simulations, which we present in the Supplement. Although the focus of this paper is not to provide a comparison of bin vs bulk schemes, we show that the differences resulting from

conversion of one ice category into another is a limitation of any scheme whether bin or bulk which uses fixed ice categories. Our results support the argument that ice phase processes may be better represented if developments in microphysics schemes starts to move away from the use of fixed ice categories.

---

## Author Comment (AC2) · 24 Dec 2016

Review of the ACP manuscript "Can models robustly represent aerosol-convection interactions if their cloud microphysics is uncertain?" by White et al. The authors used the WRF model coupled with two double-moment bulk microphysics scheme to perform cloud system-resolving simulations of convection in the Congo basin, an idealized supercell, and a case of shallow cumulus convection, and tested the sensitivity of the simulated hydrometeors and precipitation to the microphysics scheme and to CDNC perturbations. The authors showed the simulations are sensitive to microphysics parameterizations much more than CDNC, which has been showed in previous studies but this study highlighted this point to imply aerosol effects are the secondary compared with the uncertainty in cloud parameterization. They further examined the shallow cumulus convection case and found that representation of autoconversion is the dominant factor that drives differences in rain production. The paper is generally written clearly but there are confusing sentences. They are some perspectives that can not be well justified physically such as saying the aerosol-cloud interactions in their study represent the upper limit. The major problem is that their main point is based on an assumption that those twomoment schemes well represent the aerosol-cloud interactions, which is not the case based on the many past studies and a recent review paper by Khain et al 2015. Twomoment schemes have significant limitations in aerosol-cloud interaction process parameterizations such as nucleation, diffusional growth, and sedimentation, etc (detailed Khain et al. 2015). The paper did not really address the question "Can models robustly represent aerosol-convection interactions if their cloud microphysics is uncertain", so that the title needs to be changed. More literature survey is needed, especially about those studies comparing different microphysics schemes and their responses to CCN or CDNC. Those studies need to be discussed in the introduction and the relevant places in the paper, especially about some important points on the problems with the parameterizations of some specific microphysics processes in the bulk schemes. Therefore, the paper needs major revisions to be accepted as a publication in ACP.

We thank Reviewer 2 for their thoughtful and constructive comments, which we have found useful in helping us to clarify our manuscript for the reader.

We present this study as an illustration that model uncertainty in cloud impacts arising from choice of microphysics scheme can far outweigh any aerosol effects observed within a single scheme, and that this result holds for case study simulations consisting of many cloud lifecycles, for idealised simulations with open boundaries, for idealised simulations with periodic boundaries, and (by nature of the cases used in our study) across different types of convection with their inherently different response to aerosol.

Our choice to use bulk microphysics schemes is twofold; first, they remain in wide use in the community (often for practical reasons of computational cost) and even represent state-of-the-art implementation in global models, which have historically relied on much cruder microphysics representations, and second (most importantly) there is currently only one bin microphysics scheme implemented in the public version of WRF (although it comes in two slightly different versions, a 'full' and 'fast' version which differ in the number of ice categories used) and thus our study would be impossible to perform within a single modelling framework had we opted to use bin schemes.

**Specific comments:**

1. The title needs to be changed. It is relevant but the authors did not conduct an unique study to really address this question. The two-moment schemes can not robustly represent aerosol–convection interactions due to the limitations in representing the most relevant processes as detailed in Khain et al. 2015. If a bin scheme is used, you might end up with similar magnitudes of aerosol indirect effects as the differences among different microphysics schemes. In addition, for specific case simulations, one can not really reveal how it is impacted by aerosols by conducting simulations with realistic aerosols/chemistry configuration. Any aerosol properties and spatial distribution change could change cloud and precipitation.

We have updated the title to emphasize our main result: 'Uncertainty from choice of microphysics scheme in convection-permitting models significantly exceeds aerosol effects'.

Regarding the reviewer's comments on the use of a bin scheme:

We have now performed simulations using the HUJI spectral bin microphysics scheme (SBM) implemented in WRFv3.6.1 (note that this is a newer version of WRF than used in the Morrison and Thompson simulations presented in the paper). We do not wish to further extend the paper to include discussion of the impacts of bin vs bulk microphysics in these cases (because the main aim of this particular paper is to highlight within a single paper and by using a single modelling framework the large uncertainty in many case study and GCM simulations that can result from any given combination of microphysics scheme, type of convection, and given CDNC value or prescribed CCN profile). However, we now present our bin scheme results in a supplement to our paper (Figures S2, S3 and S4) because we believe our results to be of interest. We find that the uncertainty due to choice of microphysics scheme still dominates any aerosol response within each scheme in the cases we test, regardless of whether a bin or bulk scheme is used.

**2. P1, L21-22: aerosol can affect through aerosol radiative effects as well.**

We did not intend our phrasing to imply that the only effect that aerosol has on convection is an indirect effect through cloud microphysics. We have reworded the sentence now on P1 L22 – P2 L1: "One major way that aerosols can influence the properties of deep convection is through their effect on cloud microphysics."

**3. P1, L24: Albrecht, 1989 actually showed the suppression of precipitation but for warm clouds. So, the sentence is not accurate.**

We thank the reviewer for finding our omission of "warm-phase" in this sentence, and have corrected our sentence to "warm-phase precipitation", now on P2 L3.

**4. P2, L29: "Until recently" should be deleted.**

We replace "until recently" with "traditionally" (now on P5 L20), as it is only in recent years that schemes have started to move away from discrete ice categories (Morrison & Grabowski 2008, Harrington et al. 2013, Morrison & Milbrandt 2015), and we wish to make it clear to the reader that this is a relatively new and important development.

5. P4, L19-25, these sentences could be misleading. First, it is not clear what the authors mean by saying "the response of different microphysics schemes to perturbations in prescribed cloud parameters". Second, some past studies showed qualitatively different aerosol impact for different cloud types, with a purpose of illustrating a point that aerosol impacts depend on cloud type and dynamical and thermodynamic conditions of each case. So it is misleading to describe a study without giving information about cloud types or specific dynamical and thermodynamic environment. For example, the description about Fan et al. 2012 about aerosol reducing precipitation is not correct. The study did two different cloud cases over the eastern China – one deep convective cloud case with warm cloud base and weak wind shear and the other a winter stratiform cloud case, with aerosol increased precipitation for the former but reduced precipitation for the latter.

We thank the reviewer for pointing out that this was not clear to the reader.

We have heavily revised our introduction and included more detailed discussion on aerosol response in bin and bulk schemes, noting the convective regimes and large-scale environments used in each study. The paragraph P4 L19-25 has now been removed.

**6. P5, L3: cloud and precipitation responses > cloud and precipitation responses to perturbation of CDNC We thank the reviewer for noticing this omission and have amended our text (now on P8 L3) accordingly.**

7. P5, L25: Khain and Lynn 2009 is not s study with specified CDNC. CDNC is prognostic in the bin model they used.

We thank the reviewer for this correction and have amended our text.

8. P6, first paragraph, RRTM and Goddard schemes only talk hydrometeor mass from microphysics calculation (Goddard shortwave scheme takes prognostic CDNC as well). The microphysics-radiation coupling does not account for particle size changes, which means some aerosol effects are missing in those studies.

We agree with the reviewer and we also note that the microphysics-radiation coupling is only between cloud water and ice, and none of the other frozen species. This missing aerosol effect may have an especially important impact in our Congo simulations, where the Morrison scheme develops and retains significant amounts of upper-level ice, whereas the Thompson scheme converts nearly all the ice to snow (see Figures in our response to Reviewer 1), which the radiation scheme will not see. This could have significant radiative flux and feedback impacts (Thompson 2015, Atmospheric Research), which in itself originates from the use of somewhat arbitrarily defined ice categories (e.g. if the size parameter at which cloud ice is converted to snow is changed, a bulk mass of cloud ice is removed from the radiatively-coupled ice category and moved into the non-radiatively coupled snow category).

We have included these important points in our Discussion and Conclusions of the Congo basin results, P24 L6-15.

**9. Figure 3, the color scheme needs to be changed. The color difference is too little even between 0 and 100 mm.**

We have changed the colour scale of this Figure to a scale which more clearly shows the large difference in accumulated rain over the 10-day period between the models and observations. The new Figure is shown below in Figure R1.

**Figure R1**: Congo case: accumulated surface precipitation (mm) from 01 to 10 August 2007 in the Congo basin, showing data from (a) CONGO-M250, (b) CONGO-T250 and (c) observations from the TRMM 3B42 gridded 3-hourly mean merged precipitation product. The simulation data shown in this Figure has been coarsened to the 0.25 degree spatial resolution of the TRMM product.

**10. Figure 5, why compare with the climatology data? This is just a 10-day run, how should we expect it represent the climatology?**

There were so few CloudSat overpasses during the 10-day period that the resulting histogram is very noisy compared to the model data, albeit qualitativey very similar to the climatology (see figure R2 below compared to Figure 5 in the paper). This is why we originally used the observed climatology to construct Figure 5. At Reviewer 2's suggestion, we have replaced Figure 5 in our manuscript with the histogram showing 10 days of CloudSat data, Figure R2 below.

**Figure R2**: Congo case: 10-day histogram for the period 1 – 10 August 2007 of model reflectivities derived from hydrometeor fields passed through the Quickbeam radar simulator, thresholded at values greater than -20 dBZ for (a) CONGO-M250, (b) CONGO-T250, and (c) the CloudSat 2B-GEOPROF product. In (a) and (b) the models have been sampled at the times of the nearest CloudSat overpasses.

**11. P. 9 L5-7, this sentence is confusing. Need to be clarified.**

We have now expressed this sentence more clearly (P12 L22):

"The largest reflectivity values produced by the model occur in the convective region in the north of the domain, where the largest reflectivity values are detected by the satellite radar.'

**12. P. 9, L19: domain averaged cloud water in T250 is 140 times larger than M250. Something could be wrong here. Can you plot the cloudy-point average for cloud water mass total hydrometeor mass to check if they make sense? What is the maximum cloud water mass in T250 and M250, respectively?**

This is correct in the domain-average profile, because the shallow warm cloud in the M250 case precipitates out, whereas it persists in the T250 case. This leads to a significant difference in the domain-averaged cloud water mass, because there are many zero points in the M250 case which contain cloud water in T250 (see also our response, including figure, to Reviewer 2's comment 21)

Considering the cloudy-point (condensed-point) averages, we see a similar response in behaviour (increased cloud water mass in T250 compared to M250), except the magnitude is reduced because the profiles are normalized by number of points with condensed water. The condensed-point average profiles show qualitatively similar behavior in all cloud species to the domain-mean profiles (Figure R3), except cannot represent absolute changes in the number of cloudy / condensed points between the simulations.

**Figure R3**: Congo case: Cloud-and-condensed-point mean vertical profiles of difference in hydrometeor mass mixing ratios between the polluted and pristine cases, averaged over the 10 days of the Congo simulation for the Morrison scheme (left), Thompson scheme (centre) and difference between the two (right).

We note also that in our WRF-SBM simulation (Figures S2 and S3 in the supplement provided with this response) the same warm cloud mass is present and has significantly greater domain-mean values than the M250 case. We believe this difference is because, as we have shown (our Figure 16), the cloud mass in the M250 case is removed through autoconversion to rain, whereas this does not occur in the T250 case or in the SBM.

13. Figure 6, this figure need to be replotted to show differences of other hydrometeor mass clearly. Right now, only cloud water differences can be seen clearly. I would use different panels for different hydrometeors. We agree with the reviewer that this is a difficult figure to show all hydrometeor masses clearly whilst also showing the significant difference in cloud masses between the two schemes. However, we feel that using different panels for the different hydrometeors could confuse the reader, as this is not done for any other such figures. We now present the following figure, which shows all the hydrometeor classes on the same figure, as for all other such profiles in the paper, but uses a logarithmic horizontal axis even in the difference plot. This makes clear to the reader the differences of all the hydrometeor classes, not just the cloud water, whilst also showing that the difference in cloud water between the two schemes is an order of magnitude greater than between the other hydrometeor classes.

**Figure R4**: Congo case: Domain-mean vertical profiles of hydrometeor mass mixing ratios (MMR) averaged over the period 1 – 10 August 2007. (a) CONGO-M250, (b) CONGO-T250 and (c) the difference in the domain-mean hydrometeor mixing ratio profiles (CONGO-M250 minus CONGO-T250). Note the logarithmic scale used on the

horizontal axis in (a) and (b) to illustrate the large differences in the cloud water mass between the two simulations whilst also illustrating the differences in the frozen species. Note in (c) the diverging logarithmic 'difference' scale used on the horizontal axis, with 'negative' values extending from the centre to the left, and positive values extending from the centre to the right. The end of each of these axes is cut off at a value of 10-3, however these axes tend towards each other towards zero.

**14. P9, last paragraph and Figure 7, the huge increase of cloud water with the increase of CDNC with the Thompson scheme seems not reasonable. How about the change of precipitation?**

We agree with the reviewer that the huge increase of cloud water with CDNC in the Thompson scheme in the Congo basin simulations is surprising. Our results show that this is accompanied by a reduction in precipitation (Figure 7, also noted in P10 L2, continued paragraph from last paragraph on P9). This is consistent with warm-phase precipitation suppression (Albrecht 1989). This is also consistent with our warm-phase RICO results, which show an increase in domain-averaged cloud mass and reduction in precipitation with CDNC in the Thompson scheme (Figure 12).

We suggest that the development of the warm cloud mass is likely due to the background meteorological conditions these simulations are performed in (see also our response to Reviewer 2's comment 21). We have shown that the autoconversion process is responsible for removing this cloud mass in the CONGO-MORR simulations. Our Figure 14 also shows that under increased CDNC, the threshold cloud water mass required for autoconversion to begin increases in the Thompson scheme. Therefore we suggest that increased levels of CDNC in the Thompson scheme in the Congo simulations even further suppress rain formation through autoconversion.

15. P. 10, L19-21: The sentence "we also see that the simulated hydrometeor classes differ between cases: the difference in the simulated hydrometeor classes in the idealised supercell configuration is different from the difference in the real-data Congo basin configuration" is not necessary. This is what it should be since they are different convective cloud types.

We have removed this sentence from our manuscript.

**16. P11, first paragraph, the sentences in L5-6 and in L10-11 are repeated.**

The first sentence refers to SUPER-MORR and CONGO-MORR, the second sentence to SUPER-THOM and CONGO-THOM. However, we have condensed these into a single sentence at the start of the paragraph (P14 L20):

"The SUPER-MORR and SUPER-THOM cases differs qualitatively from the CONGO-MORR and CONGO-THOM cases, respectively, both in the altitudes at which the response occurs and the sign of the response of some of the hydrometeors"

17. P11, first paragraph, the main point here should be about more significant aerosol impact on hydrometeor mass on the supercell case compared with the Congo case, not the different responses of hydrometeors between the CONGO case and the supercell case, since they should be expected for completely different cases. Many past studies have showed that aerosol impacts depend on dynamics and thermodynamics of convective clouds (e.g., Khain 2009; Fan et al. 2009).

We thought it necessary to place our results by first confirming that our results reproduce that hydrometeor response differs according to cloud type. However, we agree with Reviewer 2 that it would increase the clarity of our paper if we remove such discussion before presenting our results, and thank the Reviewer for suggesting that this can be taken as assumed knowledge. We also thank the reviewer for the suggestion of making our most important point here that the significance of aerosol impact differs between cases.

We have updated our text with these changes (P15 L1).

18. P11, last paragraph: the lack of appropriate sensitivity of bulk schemes to aerosols is mainly due to the limitation of bulk scheme parameterization in nucleation, diffusional growth, and sedimentation, etc, as detailed in Khain et al. 2015. The invigoration of updrafts can not be simulated since the saturation adjustment approach for diffusional growth of droplets limit such effects. Those aspects should be considered and discussed when interpreting the results on aerosol indirect effects here. Past studies showing the limitation of bulk scheme parameterizations in representing aerosol-cloud interactions need to be surveyed and discussed.

We agree with Reviewer 2 that saturation adjustment methods can prevent important physical processes from occurring in bulk schemes. We have now made extensive note in our introduction of past studies which show the limitation of bulk schemes in representing aerosol-cloud interactions (P2 L15 through P5 L16).

We also note that some bulk schemes produce convective invigoration effects. For example, Lebo 2014 found evidence of convective invigoration under increased aerosol loading in a bulk scheme under weak shear conditions (and suppressed convection under strong shear), similar to the findings of Fan 2009 who found the

same response in a bin scheme. Seifert 2006 also found higher overshooting tops and larger sizes of cumulonimbus in a weak shear environment with increased aerosol loading.

However, we emphasise that the main focus of this paper is not an investigation into aerosol and microphysical processes of convective invigoration (of which there is an extensive body of literature), but to highlight that the uncertainty due to choice of microphysics scheme can far exceed any simulated aerosol effects (even when using the WRF-SBM bin scheme, Figure S4 in the Supplement). Our Figure 10 is mainly used to illustrate that latent heating (and therefore dynamic impacts) differences due to the choice of microphysics scheme can equal those due to different levels of CDNC in a bulk scheme. (Unfortunately we cannot provide equivalent latent heating impacts for the WRF-SBM supercell simulation presented in our Supplement because we wrote the latent heating output into the Morrison and Thompson schemes for this study and have not had the chance to do this for the version of the HUJI SBM included in the public WRF release).

We agree with Reviewer 2 that our finding should be placed in the context of the literature which discusses the ability of bulk schemes to produce invigoration effects, and we have therefore included this discussion in our text.

**19. P12, L5-7: Again physically it is supposed to be that for different types of convective cases. Hydrometeor differs and hydrometeor responses to CDNC are different as well.**

This sentence was not supposed to convey that the hydrometeors and their response to CDNC differ in different convective cases (which, as Reviewer 2 has said, is supposed to be the case), but rather to highlight the source uncertainty due to the choice of microphysics scheme: response to CDNC varies in each scheme according to cloud / convection type (known), but the difference between the response of the two schemes to CDNC across types of convection is not systematic.

We strive to make our text as clear as possible, therefore we have rephrased our text to say 'the difference between the response of the two schemes to CDNC across types of convection is not systematic' (P16 L31-32).

**20. P13, L23-25, reword the sentence. Not sure what you really want to say.**

We have removed this sentence as we would need to re-run the simulations with autoconversion and accretion rates output in order to show this point.

**21. P15, first paragraph, the authors showed the autoconversion process is not the significant process**

contributing to the large cloud mass at low –levels. Then the question is what process mainly contributes to it? We have shown that the autoconversion process is responsible for the removal of the large cloud mass at low levels in the model configuration using the Morrison microphysics scheme. We also see that this low-level liquid cloud mass forms when we run the same simulation with the WRF-SBM implementation of the HUCM microphysics (Figures S2 and S3 in the Supplement), although to a lesser extent than in the Thompson simulations, and the warm cloud produced by the HUCM SBM produces rain. We therefore suggest that it is not the Thompson scheme per se which is responsible for producing the low-level cloud mass, but rather the meteorological conditions present in which these simulations are performed.

We now discuss this in our revised manuscript, P20 L10-16.

We provide here Figure R4 showing 3D isosurfaces of cloud water and ice mixing ratios for one snapshot in time of the WRF Congo simulations, 7 days into the simulation, at 01Z. Videos of such figures show firstly that the warm cloud mass forms mostly in the south-eastern region of the domain, over the ocean, and secondly that this cloud undergoes strong diurnal forcing.

t = 145 hours (01 UTC 07 August 2007)

**Figure R4**: Three-dimensional isosurfaces of cloud ice mass (top row) and cloud water mass (bottom row) for the M250 (left column) and T250 (right column) Congo case. The isosurface shown is the  $1.E^4$  kg kg-1 surface.

However, regardless of the reasons for the development of the mass of warm cloud at low levels, the importance of our results is that under identical initial and lateral boundary meteorological conditions, the choice of microphysics representation on cloud development equals or exceeds that of cloud response to CDNC within each scheme.

**22. P15, L19-20, again, the limitation of two-moment bulk schemes in representing aerosol impacts on microphysics processes should be discussed.**

Bulk schemes have been shown to be limited in their ability to produce convective invigoration. Other studies using bulk and bin-bulk schemes have identified aerosol impacts on precipitation of up to about 15% (e.g. Kalina et al. 2014, Morrison 2012, Morrison 2011, Lebo et al. 2012, Lee & Feingold 2010, Lee & Feingold 2013, Lee 2011, van den Heever et al. 2006). Indeed, even studies using bin schemes have been shown to have little impact on total precipitation, although inducing a shift in rainfall rates (Fan et al. 2013).

We note that global modelling studies of aerosol indirect effects use bulk microphysics representations (e.g. Ghan et al. 2016, Zhang et al. 2016), and therefore our results have important implications for such studies.

We emphasise that our main result is to show that the variability due to, and within, schemes dominates any aerosol impacts on microphysics. Our results using the WRF-SBM in the idealised supercell case show that aerosol impacts in the bin scheme are of equal magnitude to those in the bulk schemes (Figure S4 in the Supplement).

We have included such a discussion in this section of the paper (P22 L24 through P23 L1) and also in the introduction (P2 L15 through P5 L16).

23. P17, L21-23, this sentence appears in a few places throughout the study, but the point can not be well justified even for aerosol indirect effects. First, you not know what the reality of aerosol look like in composition and spatial variability, many studies showed that aerosol spatial distribution could significant change storm location such as urban aerosols impact significantly on the precipitation in the downwind area of cities through aerosol indirect effects. Second, since two-moment bulk schemes even can not represent aerosol-cloud interaction processes physically, then how do you justify the aerosol impact here represent the upper limit? We have removed these statements from our manuscript.

24. P18, L29, this is definitely not the first study to consider two and more cloud cases. A thorough literature search would give you those past studies.

We have removed this entire paragraph to make the conclusions more concise.

25. P18, L30-31, again, it has been a basic understanding that hydrometeors and their responses to CCN or CDNC vary with different cloud types and convective cases.

We agree with Reviewer 2 that this is basic understanding. As Reviewer 2 feels that this can be taken as assumed knowledge, we have increased the clarity of our paper by removing all such contextual discussion.

---

## Author Comment (AC3) · 24 Dec 2016

The comment was uploaded in the form of a supplement:
http://www.atmos-chem-phys-discuss.net/acp-2016-760/acp-2016-760-AC3-supplement.pdf

---

## Author Comment (AC5) · 24 Dec 2016

**1. Impact of microphysics scheme on convective core dynamics**

We present here joint histograms for our Congo basin simulations of cloud top height in convective updraughts (columns identified with vertical velocities greater than 1 ms$^{-1}$) and the radius of the updraughts, as identified by running a connected-components labelling algorithm on the field of identified updraughts. Figure S1 shows that the most significant dynamical difference comes from choice of microphysics scheme: the Morrison scheme has a tendency towards higher frequencies of wider updraught radii with higher cloud tops than the Thompson scheme.

[Figure]

**Figure S1:** *Congo case: joint histograms of updraught radius and cloud top pressure at the top of the cloudy updraughts. MORR is shown on the top row, THOM on the bottom. CDNC values of 100, 250 and 2500 are shown in the left centre, and right, respectively. Updraughts are identified by masking points where the maximum vertical velocity exceeds 1ms$^{-1}$, and then applying a connected-components labelling algorithm to identify unique updraught areas*

**2. Performance of bin versus bulk microphysics in the Congo simulations**

We performed the 10-day Congo simulation using the HUJI spectral bin microphysics scheme (SBM, full version) implemented in WRFv3.6.1 (note that this is a newer version of WRF than used in the Morrison and Thompson simulations presented in the paper), and, when we compare the 'default' setup in each of the microphysics schemes (cloud droplet number concentration of 250 drops per cc in the Morrison scheme, 100 per cc in the Thompson scheme, and the bin scheme in its default settings) we find that the differences between the bin scheme and either of the bulk schemes are of the same order of magnitude as the differences between the two bulk schemes (Figures S2 and S3). Further, the bin scheme develops both the upper-level anvil ice that is produced in the Morrison scheme, and retains some of the low-level liquid cloud (although some of this rains out) that is persistent in the Thompson scheme but which rains out through autoconversion and subsequent accretion in the Morrison scheme (Figure S2). Therefore, the magnitude of

uncertainty introduced by the choice of microphysics scheme remains the same in this particular case whether a bin or bulk scheme is used.

[Figure]

**Figure S2**: *Congo case: Zonal-mean vertical sections of hydrometeor classes (colour contours) from 01 to 10 August 2007 for the Morrison scheme (left), Thompson scheme (centre) and SBM (right) in their default configurations. Hydrometeor mass mixing ratios are contoured at $10^{-6}$kg kg$^{-1}$. Note that to show the SBM data equivalently to the two bulk schemes, the graupel and hail categories in the SBM have been combined into the 'graupel' contour, and the ice categories (pristine ice, columns, plates and dendrites) have been combined into the 'ice' contour.*

[Figure]

**Figure S3**: *Congo case: Domain-mean vertical profiles of hydrometeor mass mixing ratios averaged over the period 01 to 10 August 2007 for the Morrison scheme (left), Thompson scheme (centre) and SBM (right) in their default configurations. Note that to show the SBM data equivalently to the two bulk schemes, the graupel and hail categories in the SBM have been combined into the 'graupel' contour, and the ice categories (pristine ice, columns, plates and dendrites) have been combined into the 'ice' contour.*

**2. Aerosol response in bin versus bulk microphysics in the idealised supercell simulations**

We could not afford the time or computational cost to run our Congo simulation with three different CCN profiles with the full SBM. Neither is the focus of this study to perform a bulk-vs-bin microphysics comparison . However, we also ran the idealised supercell case with the WRF-SBM in its default mode of continental CCN parameters, and also with maritime CCN parameters for comparison against the bulk schemes (Figure S4). We find a similar magnitude of aerosol effect whether a bin or bulk scheme is used in this particular idealised configuration, which itself is between the same order and one order of magnitude smaller than the difference due to choice of microphysics scheme in the supercell case (Figure 8 in our paper).

We note here again the finding of Kalina et al. 2014 that CCN responses are nonmonotonic. By comparing response to the same CDNC values in MORR and THOM to response to continental and maritime CCN profiles in the SBM, we enter a territory where this nonmonotonicity may introduce uncertainty to our comparison. However, we nevertheless find that even when comparing results using a bin scheme against our bulk scheme results, it is the choice of scheme which dominates the differences in the model configurations.

[Figure]

**Figure S4**: *Domain-mean vertical profiles of difference in hydrometeor mass mixing ratios between the polluted and pristine cases, averaged over the 2 hours of the supercell simulation for the Morrison scheme (left), Thompson scheme (centre) and WRF-SBM (right).*

---

## Referee Report (RR1)

**Review of the paper "Uncertainty from choice of microphysics scheme in convection-permitting models significantly exceeds aerosol effects",
authored by Bethan White, Edward Gryspeerdt, Philip Stier, Hugh Morrison, Gregory Thompson, and Zak Kipling**

This study is a revised version of the paper "Can models robustly represent aerosol–convection interactions if their cloud microphysics is uncertain?", authored by B. White, E. Gryspeerdt, P. Stier, H. Morrison, and G. Thompson. Two widely used bulk parameterization schemes referred to as "MORR" (Morrison and Milbrandt (2011) and "THOM" (Thompson et al. (2004, 2008)) were tested by simulations of three case studies with different type of convection: from shallow convection to a supercell storm. The simulations were performed at a priory given droplet concentrations of 100 cm-3, 250 cm-3 and 2500 cm-3.

A huge difference in cloud microphysical structure simulated by these two schemes is reported.

Both bulk schemes turned out to be insensitive to droplet concentration. So, the difference in results are related to the differences in the bulk-parameterization schemes.

The paper was improved in course of the revision. The overview of the state-of-the art bulk schemes includes now many necessary references.

In my view, the most interesting section in the revised paper is the analysis of the reasons of differences between the results obtained using these two. The authors stress a crucial role of representation of autoconversion process. The Thompson scheme uses one of versions the Berry and Reinhard (1974) autoconversion scheme. Gilmore and Straka (2008) (this study is referred now in the new article version) showed that the rates of autoconversion predicted by different versions of the Berry and Reinhard (1974) scheme differ by orders of magnitude. So, a justification of the choice of particular autoconversion scheme is required.

The Morrison scheme uses parameterization of autoconversion developed by Khairoutdinov and Kogan (2000) for drizzle formation in marine stratocumulus. Note that the mechanism of drizzle formation in Sc substantially differs from raindrop formation in Cu and, of course, in deep convective clouds. In this relation, Khairoutdinov and Kogan (2000) wrote in their article: 1) "The proposed bulk microphysical parameterization has been developed and tested for thermodynamic conditions typical for the midlatitude and extratropical stratocumulus layers formed over the areas of upwelling off the west coasts of continents; therefore, it may not be valid to extrapolate its use to other cloud types and conditions" and 2) "We have to emphasize that the proposed scheme is

intended for LES of convective STBL with a spatial resolution of tens of meters. Such an LES resolves most eddies of turbulent flow and, consequently, spatial variation in supersaturation, water content, cloud condensation nuclei (CCN) count, drop concentration, etc. This auxiliary information enables one to add a level of complexity to the traditional bulk microphysics schemes by adding, for example, the explicit CCN–cloud drop concentration feedback, as done in this study. Therefore, the proposed scheme cannot be simply extrapolated for use in larger-scale models since the derived water conversion rates depend *nonlinearly* on local (eddy scale) cloud variables".

So, on one hand it is good that the important reason of the differences between results of the Thompson and of the Morrison schemes is found. On the other hand, a justification and reasoning of utilization of the Khairoutdinov and Kogan (2000) parameterization for conditions quite different from those in Sc are required.

3. The authors illustrate vertical profiles of mass contents of different hydrometeors averaged over the entire computational area. As a result, all information concerning the microphysical structure of clouds simulated by different schemes turns out to be lost, at least for specialists in cloud microphysics. It is necessary to present vertical profiles of maximum values of the mass contents. The profiles of cloud averaged values would be also useful. These figures should be accompanied by corresponding comments and analysis.

4. I still do not understand the reason of the existence of small cloud droplets near the surface (Fig. 6 and Fig. 7). If spontaneous breakup of raindrops is not included, the reasons of this very strange effect should be explained. What is relative humidity in the BL?

5. The section of Conclusions was improved, but still remains weak. The key finding as it is formulated in the paper is: "In the context of our finding, this strongly suggests that an accurate description of the autoconversion process in warm-rain regimes is fundamental not only to a realistic representation of cloud and precipitation, but also to its response to varying aerosol concentration".

This finding is not new and seems somehow trivial. It is not necessary to analyze in detail three case studies to conclude that the level of raindrop formation and the growth rate of raindrop mass are of crucial importance for warm and mixed-phase cloud microphysics. Actually, autoconversion rate determines the difference between cloud types: maritime vs. continental.

No solution or even suggestion concerning the ways to improve the representation of the autoconversion is proposed in the study. At the same time two moment bulk schemes allow

calculation of the mean volume radius. Note that the mean volume radius is a very robust quantity, which vertical profile depends on droplet concentration, i.e on the CCN concentration. The mean volume (or effective) radius can be also calculated using adiabatic LWC and droplet concentration. It is also known that raindrop onset begins when the mean volume radius exceeds its a critical value of 13-14 um (Freud and Rosenfeld, 2012, Khain et al., 2013, Rosenfeld et al. 2014). This allows to calculate the height of the first raindrop formation quite accurately. Might be this condition can be used for testing and improvement of the schemes?

I also recommend to refer the recent studies by Igel and van den Heever (2016a,b, 2017), where different values of the shape parameters of Gamma distribution is considered and important reason of difference between the results of bulk schemes. Igel and van den Heever proposed the optimum shape parameter of gamma distribution at the stage of diffusion growth.

Of course, available bin microphysics models (WRF, SAM, parcel models with a very detailed description of raindrop formation) can be useful.

The discussion of the possible ways to improve the autoconversion schemes is desirable.

I recommend to accept the paper with *major revision*.

References:

Freud, E., and D. Rosenfeld (2012), Linear relation between convective cloud drop number concentration and depth for rain initiation, J. Geophys. Res., 117, D02207, doi:10.1029/2011JD016457

Igel, A.L. and S.C. van den Heever, 2016a: The importance of the shape of cloud droplet size distributions in shallow cumulus clouds. Part I: Bin microphysics simulations. Accepted pending revision at *J. Atmos. Sci.*

Igel, A.L. and S.C. van den Heever, 2016b: The importance of the shape of cloud droplet size distributions in shallow cumulus clouds. Part II: Bulk microphysics simulations. Accepted pending revision at *J. Atmos. Sci.*

Igel A. L. and Susan C. van den Heever (2017) The Role of the Gamma Function Shape Parameter in Determining Differences between Condensation Rates in Bin and Bulk Microphysics Schemes

Khain, A., T. V. Prabha, N. Benmoshe, G. Pandithurai, and M. Ovchinnikov, 2013: The mechanism of first raindrops formation in deep convective clouds, *J. Geophys. Res. Atmos.*, **118**, 9123–9140, doi:10.1002/jgrd.50641

Rosenfeld, D., B. Fischman, Y. Zheng, T. Goren, and D. Giguzin (2014), Combined satellite

and radar retrievals of drop concentration and CCN at convective cloud base, *Geophys. Res. Lett.*, *41*(9), 3259–3265, doi:10.1002/2014GL059453.

---

## Referee Report (RR2)

**Review of the paper "Uncertainty from choice of microphysics scheme in convection-permitting models significantly exceeds aerosol effects",**
**authored by Bethan White, Edward Gryspeerdt, Philip Stier, Hugh Morrison, Gregory Thompson, and Zak Kipling**

The paper was improved in course of the revision.

I partially satisfied with the response of the authors.

Nevertheless, some comments remain unanswered.

For instance, both bulk schemes turned out to be insensitive to droplet concentration. Can the author comment this insensitivity? Do they consider this insensitivity as a natural property of cloud systems simulated in the study, or they attribute this insensitivity to the specific features of the bulk schemes tested in the study? Corresponding discussion should be included into the Conclusion Section.

In the revised paper the authors presented more detailed discussion of the bulk schemes used. Note that the fact that many scientists use for deep convection simulations the autoconversion scheme developed for slightly drizzling stratiform clouds does not indicate yet the ability of the scheme to simulate deep convection well.

Which scheme (Berry and Reinhard , 1974 or of Khairoutdinov and Kogan (2000)) show better results? Which scheme is recommended by the authors?

These comments and remarks are minor. So, I agree with publication of the paper with *minor* revisions.

---

## Author Response (AR2)

**RESPONSE TO REVIEWER 1:**

I agree to have the paper accepted by ACP. The only comment that I have is that the authors might want to check the aerosol concentrations in the WRF-SBM simulations that they added in the supplemental material to make sure the simulations were done correctly, because there is a bug in aerosol setup in the SBM version in WRF3.6.1.

Thank you for highlighting this potential technical issue. We have checked our supercell SBM results against a newer version of the SBM in WRF (WERFv3.7.1) and our results and conclusions are not changed.

**RESPONSE TO REVIEWER 2:**

**Review of the paper "Uncertainty from choice of microphysics scheme in convection-permitting models significantly exceeds aerosol effects",**
**authored by Bethan White, Edward Gryspeerdt, Philip Stier, Hugh Morrison, Gregory Thompson, and Zak Kipling**

1. In my view, the most interesting section in the revised paper is the analysis of the reasons of differences between the results obtained using these two. The authors stress a crucial role of representation of autoconversion process. The Thompson scheme uses one of versions the Berry and Reinhard (1974) autoconversion scheme. Gilmore and Straka (2008) (this study is referred now in the new article version) showed that the rates of autoconversion predicted by different versions of the Berry and Reinhard (1974) scheme differ by orders of magnitude. So, a justification of the choice of particular autoconversion scheme is required.

The reviewer is correct that Gilmore and Straka very nicely illustrated a very large sensitivity of autoconversion results dependent upon how the subsequent authors implemented the exact details of Berry and Reinhardt (1974; hereafter B&R74). This is quite illustrative of the problems as one person or another does or does not follow the identical details as found in the original research. This is also well highlighted in Thompson et al. (2004): note in particular the reference to Walko et al. (1995) in the footnote at the bottom of page 521. We justify our choice of staying with B&R74 by stating that we have compared its results as in Thompson et al (2004 and 2008) against a bin/explicit microphysics scheme of Geresdi (1998) and found very favorable comparisons. One coauthor of the current manuscript (G. Thompson) has extensively compared various autoconversion schemes, including one that gained widespread usage in recent years by Khairoutdinov and Kogan (2000) and still believes the choice of B&R74 to be superior via comparisons to observations in real-time numerical weather prediction models.

To make this justification clear to the reader, we have made note of this in our revised manuscript in the Section 3.4 where the autoconversion representations are introduced, at Page 18, Lines 26 – 32.

2. The Morrison scheme uses parameterization of autoconversion developed by Khairoutdinov and Kogan (2000) for drizzle formation in marine stratocumulus. Note that the mechanism of drizzle formation in Sc substantially differs from raindrop formation in Cu and, of course, in deep convective clouds. In this relation, Khairoutdinov and Kogan (2000) wrote in their article: 1) "The proposed bulk microphysical parameterization has been developed and tested for thermodynamic conditions typical for the midlatitude and extratropical stratocumulus layers formed over the areas of upwelling off the west coasts of continents; therefore, it may not be valid to extrapolate its use to other cloud types and conditions" and 2) "We have to emphasize that the proposed scheme is intended for LES of convective STBL with a spatial resolution of tens of meters. Such an LES resolves most eddies of turbulent flow and, consequently, spatial variation in supersaturation, water content, cloud condensation nuclei (CCN) count, drop concentration, etc. This auxiliary information enables one to add a level of complexity to the traditional bulk microphysics schemes by adding, for example, the explicit CCN–cloud drop concentration feedback, as done in this study. Therefore, the proposed scheme cannot be simply extrapolated for use in larger-scale models since the derived water conversion rates depend *nonlinearly* on local (eddy scale) cloud variables".

So, on one hand it is good that the important reason of the differences between results of the Thompson and of the Morrison schemes is found. On the other hand, a justification and reasoning of utilization of the Khairoutdinov and Kogan (2000) parameterization for conditions quite different from those in Sc are required.

The reviewer is correct that the Khairoutdinov and Kogan (2000; hereafter KK2000) scheme was initially developed and applied for LES of stratocumulus. This has motivated the implementation of additional options besides KK2000 for autoconversion and accretion in newer schemes such as P3 (Morrison & Milbrandt 2015a,b). However, we note that other than varying the prescribed values of cloud droplet number concentrations we are running the schemes in their baseline configurations as available in the main WRF release. Thus, although we are not advocating the use of KK2000 for non-stratocumulus cases, the case we present is the same configuration used by any other WRF user running deep convection simulations with the Morrison scheme.

Secondly, we agree with the reviewer that virtually all physically-based autoconversion/accretion schemes suffer from the issue of spatial resolution and not resolving local variations in water content and concentration, since they are typically based on bin or numerical model calculations of the growth of drops by collision-coalescence based on local water contents and concentrations. This brings up the larger issue of the effects of sub-grid scale cloud water variability on microphysical process rates. This is of critical importance in large scale models, and has been addressed by coupling schemes like KK2000 with a sub-grid scale pdf representation of cloud water (e.g., Morrison and Gettelman 2008, J. Climate). On the other hand, the effects of sub-grid scale cloud variability on grid-mean autoconversion and accretion is less clear for models at convection-permitting scales, although nearly all models neglect the effects of sub-grid cloud water variability at these scales. While the authors recognize the potential importance of grid resolution sensitivities at these scales, due not just to cloud water variability but especially sensitivity of the cloud/convective dynamics, we note that we are running the model and microphysics schemes in the typical setup for a convection-permitting model (that is, neglecting sub-grid cloud variability).

We note that one of the main aims of our study is in fact to highlight the uncertainty in commonly used model configurations, which are exactly based on these schemes.

We have made note of this justification in our revised manuscript, at Page 19, lines 6 – 9.

3. The authors illustrate vertical profiles of mass contents of different hydrometeors averaged over the entire computational area. As a result, all information concerning the microphysical structure of clouds simulated by different schemes turns out to be lost, at least for specialists in cloud microphysics. It is necessary to present vertical profiles of maximum values of the mass contents. The profiles of cloud averaged values would be also useful. These figures should be accompanied by corresponding comments and analysis.

We note that in the first revision of the paper we updated all profile figures to present both domain-averaged cloudy column profiles, as included in our original manuscript, and cloud-only averaged profiles as requested by the reviewer in round 1 of the review. This was accompanied by corresponding comments and analysis.

As noted in our previous response to the reviewer, the relevant Figures are Figures 6,7,8,9,11,12 and an example of the condensed-point (cloud and precip) average profiles was provided in our response to Reviewer 2's comment 12 in the first set of responses to the reviewers.

However, we do not agree that showing domain-maximum values is the best way for comparison of the cases we present. Although such analysis is useful to understand the evolution of a single cloud, such maxima in our simulations would be calculated over many cloud types and regions in the Congo case, and over multiple cloud lifecycles in both the Congo case and the RICO case. Maximum values of the mass contents represent just one point in the entire domain and can lead to improper conclusions, especially in our Congo simulation where multiple cloud types exist in the same domain but do not necessarily interact.

4. I still do not understand the reason of the existence of small cloud droplets near the surface (Fig. 6 and Fig. 7). If spontaneous breakup of raindrops is not included, the reasons of this very strange effect should be explained. What is relative humidity in the BL?

We note that the profiles in Figures 6 and 7 show the mass mixing ratios, not number concentrations, of cloud water contents (cloud droplet numbers are prescribed in both the Morrison and Thompson schemes). Therefore, the profiles show a large amount of liquid cloud mass in the Thompson scheme, which we have shown to rain out in the Morrison scheme (the authors note that raindrop breakup is included in both the Morrison and Thompson schemes, following the form in Verlinde & Cotton but implemented slightly differently using different diameter thresholds, different self-collection efficiencies etc).

We showed in our first response to the reviewer that this is low-level warm cloud that forms over the ocean. Since both sets of simulations are driven by the same initial and boundary conditions, this could be due to a moist bias in the reanalysis in this region (Washington 2013). However, in a region such as the Congo basin the presence of significant low-level moisture is not at all surprising. Further, it is likely that some of the low cloud formation is driven by saturation from rain evaporation.

Figure R1 shows the mean relative humidity profile for all sets of simulations. Mean values (all domain, all simulation) of relative humidity in the BL is between 55 and 60%, with values higher in simulations using the Morrison scheme. That low-level relative humidity in the (precipitating warm cloud) Morrison simulations is greater than that in the (non-precipitating warm cloud) Thompson simulations, when both are driven by the same BCs, strongly suggests the effect of rain evaporation increasing low-level humidity in the Morrison simulations.

[Figure]

**Figure R1:** *Congo case: domain mean profiles of relative humidity over the 10 days of the simulation for the Morrison (solid lines) and Thompson (dashed lines) schemes, for simulations with prescribed cloud droplet number concentrations of 100 (black lines), 250 (blue lines) and 2500 (red lines) drops per cc.*

5. The section of Conclusions was improved, but still remains weak. The key finding as it is formulated in the paper is: "In the context of our finding, this strongly suggests that an accurate description of the autoconversion process in warm-rain regimes is fundamental not only to a realistic representation of cloud and precipitation, but also to its response to varying aerosol concentration". This finding is not new and seems somehow trivial. It is not necessary to analyze in detail three case studies to conclude that the level of raindrop formation and the growth rate of raindrop mass are of crucial importance for warm and mixed-phase cloud microphysics. Actually, autoconversion rate determines the difference between cloud types: maritime vs. continental. No solution or even suggestion concerning the ways to improve the representation of the autoconversion is proposed in the study. At the same time two moment bulk schemes allow calculation of the mean volume radius. Note that the mean volume radius is a very robust quantity, which vertical profile

depends on droplet concentration, i.e on the CCN concentration. The mean volume (or effective) radius can be also calculated using adiabatic LWC and droplet concentration. It is also known that raindrop onset begins when the mean volume radius exceeds its a critical value of 13-14 um (Freud and Rosenfeld, 2012, Khain et al., 2013, Rosenfeld et al. 2014). This allows to calculate the height of the first raindrop formation quite accurately. Might be this condition can be used for testing and improvement of the schemes?

We note that the finding quoted by the reviewer is not our key finding. Indeed, our key result is presented as 'variabilty in aerosol response due to choice of microphysics scheme differs not just between schemes, but that the inter-scheme variability differs between cases of convection'.

We agree with the reviewer that for warm clouds, autoconversion should be the main way in which differences in droplet concentration affect simulations (as opposed to accretion, whose dependence on number concentration is not even included in KK2000, or droplet sedimentation, or effects on the droplet size distribution shape parameter that depends on the number concentration in both the Morrison and Thompson scheme used in this study). However, we note that the paper is already long and contains a lot of analysis, and whilst further testing of the autoconversion process between the two schemes would be interesting it is not the focus of this paper. Furthermore, the Thompson scheme's implementation of B&R74 contains the principle point that the reviewer makes: that collision-coalescence produced warm rain begins at almost exactly 14 microns using the method as implemented by Thompson. This is one of the principle reasons B&R74 was chosen rather than K&K2000. While other persons who claim to implement B&R74 (as pointed out by Gilmore and Straka), they have often done so not using the exact same 3 characteristic diameters of B&R74's original paper; whereas Thompson has.

We make note of this justification of the B&R74 implementation in page 18, lines 26 – 32 of our revised manuscript.

We have included in our discussion a note that care should be applied using autoconversion schemes in different regimes for which they were originally developed, as in the KK2000 scheme. This can be found on Page 26, lines 17 – 19 of the revised manuscript.

We note that cloud droplet number concentration is prescribed as a parameter in both bulk schemes used in this paper (i.e. neither version of the scheme used here is prognostic in cloud droplet number concentration). Therefore, for the K&K2000 scheme calculation of the mean volume radii based on the specified number concentrations is unlikely to provide much meaningful information (note that, as described above, B&R74 does effectively calculate a volume-based mean radius). However, we have included a description of the reviewer's suggested method to calculate the effective radius for readers who may be interested to perform such tests in schemes with prognostic cloud droplet number concentrations. This can be found on Page 26, lines 29 – 31 of the revised manuscript.

However, we emphasize again that the focus of the paper is not to make improvements to each of the schemes, but to highlight the wide variability in response of these two bulk microphysics schemes to aerosol not just with respect to each other, but more importantly with respect to each other between types of convection – a result that can likely be extended to all bulk schemes and therefore to global models as well as cloud-resolving models.

I also recommend to refer the recent studies by Igel and van den Heever (2016a,b, 2017), where different values of the shape parameters of Gamma distribution is considered and important reason of difference between the results of bulk schemes. Igel and van den Heever proposed the optimum shape parameter of gamma distribution at the stage of diffusion growth.

The suggested references are now included in our revised manuscript on Page 3, lines 24 – 33.

Of course, available bin microphysics models (WRF, SAM, parcel models with a very detailed description of raindrop formation) can be useful. The discussion of the possible ways to improve the autoconversion schemes is desirable.

We have included the reviewer's suggestion of ways to test and improve schemes on Page 26, lines 32 – 33 of the revised manuscript.

---

## Author Response (AR3)

The paper was improved in course of the revision. I partially satisfied with the response of the authors. Nevertheless, some comments remain unanswered.

For instance, both bulk schemes turned out to be insensitive to droplet concentration. Can the author comment this insensitivity? Do they consider this insensitivity as a natural property of cloud systems simulated in the study, or they attribute this insensitivity to the specific features of the bulk schemes tested in the study? Corresponding discussion should be included into the Conclusion Section.

We highlight to the reviewer that the bulk schemes are not insensitive to droplet concentration. Yes, the magnitude of the responses are small when averaged over the cloud systems and lifecycles, but there is a response and indeed the major point of our paper is that this response is highly variable under different conditions.

Moreover, we have shown that the magnitude of the response to (proxies for) aerosol in the idealised supercell case is the same as when we use the SBM bin scheme in WRF as for the two bulk schemes used in our main study. This is shown in Figure S4 in our supplement, provided with revision 1 of our paper, and was already discussed in our previous revised manuscript on page 16, line 1.

We cannot make any comment on the comparison of the pathways leading to this response in the two bulk schemes and the bin scheme. Although we agree with the reviewer that this would provide interesting insight to the bulk schemes used and the cloud systems simulated in this study, this would be far beyond the scope of the current paper, which aims to highlight the variable and uncertain response in the bulk schemes tested.

In the revised paper the authors presented more detailed discussion of the bulk schemes used. Note that the fact that many scientists use for deep convection simulations the autoconversion scheme developed for slightly drizzling stratiform clouds does not indicate yet the ability of the scheme to simulate deep convection well. Which scheme (Berry and Reinhard , 1974 or of Khairoutdinov and Kogan (2000)) show better results? Which scheme is recommended by the authors?

We agree with the Reviewer that we cannot directly advocate the use of the autoconversion schemes implemented in the two bulk schemes for studies of deep convection. This was already noted and discussed in revision 2 of our paper, both in page 19 paragraph 1, and also in our conclusions (page 26 lines 27 – 29).

However, the point of the paper is not to test autoconversion schemes and it is beyond the scope of the paper to test and recommend one autoconversion scheme (BR74 or KK2000) over the other. This would be a complete study in itself. Indeed, we would not be justified to claim one scheme as better than the other from the sets of still limited cases used in our study.

Moreover, because there are so many competing processes besides autoconversion, including a number of microphysical and dynamical processes, it could be misleading to claim that one scheme is better than the other just based on bulk comparison with observations from a few cases. It would make more sense to do off-line testing of

autoconversion schemes based on detailed in-situ observations and calculations, as was done by e.g. Wood (2005), who tested KK2000. However, we note again that this is far beyond the scope of our paper and would be a complete study in itself.

We have added the following paragraph to the conclusions in our revised paper on page 26, line 35 through page 27 line 6:

"Based on the limited set of cases in our study, we would not be justified in recommending one of the autoconversion schemes over the other. Moreover, because there are so many competing processes besides autoconversion, including a number of microphysical and dynamical processes, it could be misleading to claim that one scheme is better than the other just based on bulk comparison with observations from a few cases. For those interested in testing and evaluating the autoconversion schemes, we suggest that the best approach would be to perform off-line testing based on detailed in-situ observations and calculations, as was done by e.g. Wood (2005), who tested the Khairoutdinov and Kogan (2000) autoconversion scheme in such a manner."

**References:**

Wood, R., 2005: Drizzle in stratiform boundary layer clouds, Part II: Microphysical aspects. J. Atmos. Sci., 62, 3034.